# FROM FRAMES TO SEQUENCES: TEMPORALLY CONSISTENT HUMAN-CENTRIC DENSE PREDICTION

## ABSTRACT

In this work, we focus on the challenge of temporally consistent human-centric dense prediction across video sequences. While progress has been made in per-frame predictions of depth, surface normals, and segmentation, achieving stability under motion, occlusion, and illumination changes remains difficult. For this, we design a synthetic data pipeline that produces large-scale photorealistic human images and motion-aligned video sequences with high-fidelity annotations. Unlike prior static data synthetic pipelines, our pipeline provides both frame-level and sequence-level supervision, supporting the learning of spatial accuracy and temporal stability. Building on this, we introduce a model that integrates human-centric priors and temporal modules to jointly estimate temporally consistent segmentation, depth, and surface normals within a single framework. Our two-stage training strategy, combining static pretraining with dynamic sequence supervision, enables the model to first acquire robust spatial representations and then refine temporal consistency across motion-aligned sequences. Extensive experiments show that we achieve state-of-the-art performance on THuman2.1 and Hi4D and generalize effectively to in-the-wild videos.

## 1 INTRODUCTION

In recent years, human-centric vision has advanced in both 2D and 3D applications (Xiu et al., 2022; Weng et al., 2022; Zhang et al., 2023; Hu, 2024; Khirodkar et al., 2024; Drobyshev et al., 2022; Zhang et al., 2019; Lin et al., 2014). Current methods can estimate human pose (Cao et al., 2017), and predict dense maps such as depth (Khirodkar et al., 2024; Saleh et al., 2025) and surface normals (Khirodkar et al., 2024; Saleh et al., 2025; Saito et al., 2020; Xiu et al., 2023). Despite recent progress, achieving accurate and temporally consistent predictions in unconstrained videos remains difficult. The main challenges are: (i) the lack of large-scale human-centric video data with paired annotations for dense predictions such as depth, surface normals, and segmentation masks; and (ii) the difficulty for models to simultaneously achieve temporal stability and multi-task learning.

More recently, several methods have shown strong single-image results in estimating depth, surface normals, and segmentation masks from a single image. However, most of these approaches remain optimized for per-frame accuracy and rarely introduce explicit temporal constraints when applied to video. As a result, their predictions often suffer from temporal inconsistency, manifesting as flickering or abrupt discontinuities across frames. For instance, DAViD (Saleh et al., 2025) uses post-processing to mitigate flickering, but artifacts persist under fast motion, occlusion, and lighting changes. VDA (Chen et al., 2025) achieves temporally consistent depth estimation, due to it is trained on general-purpose datasets, it struggles to reconstruct fine-grained human geometry, including hair strands or clothing wrinkles. Jointly predicting depth and surface normals is also challenging. Although these presentations are geometrically related, their supervision emphasizes different spatial scales, which can destabilize shared representations in multi-task learning. Furthermore, current models are typically trained without human-centric priors, which leads to limited modeling of human structure. Finally, the absence of paired human video annotations that simultaneously provide segmentation masks, depth, and surface normals makes it difficult to learn shared features that generalize reliably across tasks.

In this work, we address these issues from both data and modeling perspectives to achieve temporally consistent and multi-task human-centric dense prediction. We propose a human-centric data synthe-

sis pipeline that generates photorealistic images with high-fidelity ground-truth annotations. Beyond static renderings, we incorporate AMASS (Mahmood et al., 2019) to produce dynamic sequences with motion-aligned temporal annotations. Each synthesized sample provides static RGB frames with masks, depth, and surface normals, together with dynamic sequences for temporal supervision. Unlike prior works such as Sapiens (Khirodkar et al., 2024) and DAViD (Saleh et al., 2025), which primarily scale data on generic architectures, we design a model that explicitly leverages human-centric priors (*i.e.*, CSE (Neverova et al., 2020)). Our model supports multiple temporally consistent dense prediction tasks, including segmentation, depth, and surface normals, within a single architecture and without task-specific fine-tuning. Trained solely on synthetic data, it achieves state-of-the-art results across benchmarks and generalizes effectively to in-the-wild human images and videos. Our contributions are summarized as follows.

- We build a scalable data synthesis pipeline for human-centric frames and videos with pixel-accurate depth, normals, and segmentation. We will release it to support community research on temporal consistency and multi-task learning.
- We introduce a ViT-based architecture that integrates human geometry priors to jointly predict temporally consistent segmentation, depth, and surface normals.
- To alleviate artifacts arising from feature fusion, we propose an adaptive channel re-weighting module that enhances the reliability of geometry representations.
- The method achieves state-of-the-art results on THuman2.1 and Hi4D for both depth and surface normal estimation, and transfers well to in-the-wild videos.

## 2 RELATED WORK

### 2.1 HUMAN VISION DATA

Recent progress in computer vision largely depends on the availability of high-quality training data (Yang et al., 2024a;b; Siméoni et al., 2025; Miao et al., 2025), and this also applies to human-centric applications (Khirodkar et al., 2024). Tasks such as face detection (Viola & Jones, 2004), pose estimation (Andriluka et al., 2014), landmark localization (Zhu & Ramanan, 2012), and semantic segmentation (Kirillov et al., 2023) rely on existing annotation tools. In contrast, dense prediction tasks such as depth (Wang et al., 2025a) and surface normal (Ye et al., 2024) estimation remain difficult to annotate manually. To address this challenge, several works use multi-view capture (Yin et al., 2023b; Yu et al., 2021; Martinez et al., 2024) to reconstruct human meshes. These datasets provide useful supervision, but they show limited subject and scene diversity due to high acquisition costs, and they often lose fine-scale details because they rely on model fitting or photogrammetry. More recently, DAViD (Saleh et al., 2025) combines data generation strategies with updated facial models to produce realistic human datasets with precise ground-truth annotations. However, even with large-scale datasets, most of these data are static, and data for dense dynamic prediction is still scarce. Our data synthesis pipeline directly targets this gap and enables high-fidelity synthesis for dynamic scenarios.

### 2.2 HUMAN VISION TASK

Early research focused primarily on tasks such as human keypoint estimation (Chen et al., 2018b; Fang et al., 2017; Huang et al., 2017; Khirodkar et al., 2021; Newell et al., 2016; Papandreou et al., 2017; Sun et al., 2019; Xiao et al., 2018) and body-part segmentation (Xia et al., 2017; 2016; Luo et al., 2018; Gong et al., 2018; 2017; Fang et al., 2018). Representative methods such as OpenPose (Cao et al., 2019) tackled multi-person 2D pose estimation. By jointly modeling body, hand, and facial joints, they achieved strong performance in pose and part detection on static images. Recent work has expanded to broader dense prediction tasks beyond keypoints and segmentation, including depth estimation (Bhat et al., 2023; Yin et al., 2023a; Jafarian & Park, 2021; Birkl et al., 2023) and surface normal prediction (Eigen & Fergus, 2015; Ladický et al., 2014; Saito et al., 2020; Xiu et al., 2023). For example, Sapiens (Khirodkar et al., 2024) leverages large-scale in-the-wild human images for pre-training and fine-tuning on 2D pose estimation, part segmentation, depth, and normal prediction, showing strong generalization to natural scenes. DAViD (Saleh et al., 2025) further achieves competitive results by fine-tuning DINOv2 (Oquab et al., 2023) on synthetic data. Despite their broad task coverage, these methods remain limited in stability when applied to dynamic

video scenes. In this work, we go beyond static image training by introducing supervisory signals from video sequences. This improves stability under motion, occlusion, and illumination variations, leading to more robust and generalizable predictions in natural scene videos.

## 2.3 DENSE PREDICTION ARCHITECTURES

Dense prediction has transitioned from CNN encoder–decoder baselines (Ronneberger et al., 2015; Chen et al., 2018a) with skip connections to transformer backbones trained with strong pretraining and scalable supervision. DPT (Ranftl et al., 2021) shows that a ViT encoder (Dosovitskiy et al., 2020) with a lightweight convolutional decoder yields fine-grained and globally consistent outputs for depth and segmentation, and it generalizes well across datasets. Large-scale self-supervised pretraining further improves transfer, and features from DINOv2 (Oquab et al., 2023) are widely used as a shared backbone for dense tasks without heavy task-specific heads. The Depth Anything family, especially Depth Anything V2 (Yang et al., 2024b), scales supervision using a stronger synthetic teacher and large pseudo-labeled collections of real images. The models span tens of millions to over one billion parameters and achieve improved accuracy and speed. Marigold (Ke et al., 2024) adapts a pretrained latent diffusion model to monocular depth with lightweight fine-tuning on synthetic data and reports strong cross-dataset results. For human-centric dense estimation, Sapiens (Khirodkar et al., 2024) uses a ViT backbone with lightweight task heads. In contrast, DAViD (Saleh et al., 2025) employs a dual-branch design with a ViT encoder branch and a shallow fully convolutional branch, and the features are fused in a DPT-style decoder before lightweight heads. We propose a model that injects explicit human priors into the backbone to encode body topology and part correspondence, which improves human-centric dense prediction.

## 3 METHODOLOGY

### 3.1 HUMAN-CENTRIC SYNTHETIC DATA PIPELINE

Our data synthesis pipeline consists of two stages: composition and rendering.

**Composition stage.** We leverage some character generation software (i.e., DAZ 3D[1], MakeHuman[2], Character Creator[3]) to compose clothed human models. Assets are divided into four categories—body, top, bottom, and shoes—so that they can be sampled independently. We randomize body shape and pair tops, bottoms, and shoes to generate diverse outfits. This independent sampling strategy increases the coverage of outfit combinations without requiring manual curation. For asset textures, we apply three categories of augmentations to the diffuse maps. The first introduces appearance variations through hue adjustment, per-channel intensity scaling, and low-magnitude noise. The second generates uniform solid-color textures to diversify simple surface representations. The third replaces textures using external resources, including the Describable Textures Dataset (Cimpoi et al., 2014), the ALOT dataset (Burghouts & Geusebroek, 2009), and an internal texture collection. For these replacements, we apply preprocessing operations such as resizing, tiling, mirrored tiling, and HSV-based recoloring to accommodate both colored and grayscale inputs. In total, we compose about 200K unique identities for rendering.

**Rendering stage.** We import the composed models into Blender[4] to render RGB images, depth maps, surface normal maps, and segmentation masks, from which we generate both static and dynamic data (In Figure 1). For image data, we follow the Sapiens protocol (Khirodkar et al., 2024) by randomly sampling camera viewpoints to render three perspectives: face, upper body, and full body. For video data, we animate the models with motion capture sequences from the AMASS dataset (Mahmood et al., 2019), which provides skeletal trajectories. We exclude sequences with poses such as lying and uniformly sample up to 500 frames per sequence. Each model is paired with a randomly selected trajectory. To further increase diversity, we randomize the camera focal length, enable subject-tracking, and apply camera rotations around the animated model during rendering.

---

[1] https://www.daz3d.com/
[2] http://www.makehumancommunity.org/
[3] https://www.reallusion.com/character-creator/
[4] https://www.blender.org/

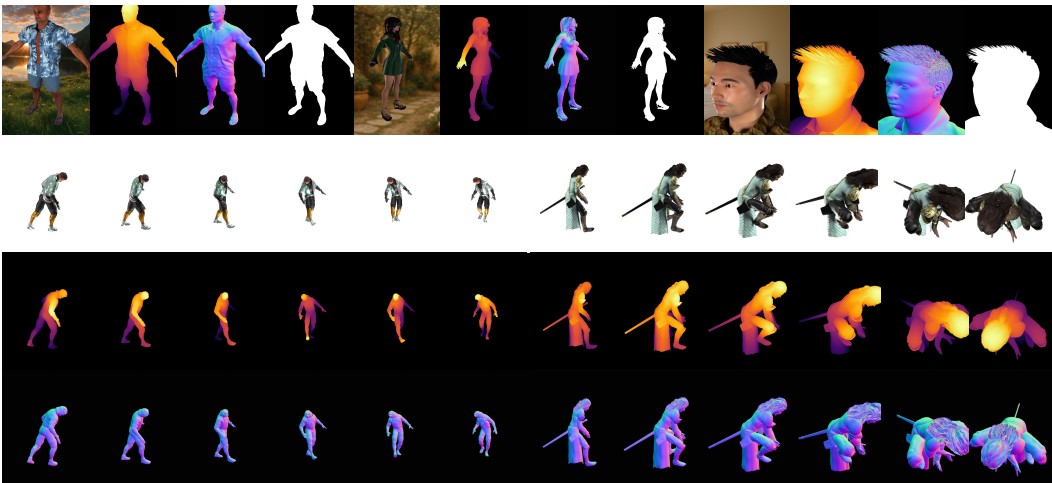

Figure 1: **Synthesis Data sample.** Ground-truth synthetic annotations of depth, surface normals, and masks for image and video sample data.

## 3.2 MODEL ARCHITECTURE

We build upon the recent paradigm of ViT-based dense prediction and adapt it to human-centric tasks. While approaches such as Sapiens (Khirodkar et al., 2024) and DAViD (Saleh et al., 2025) achieve strong performance, they remain largely task-agnostic and do not explicitly incorporate human geometry. Our objective is to introduce human geometric priors into the representation learning process, enabling a framework for human-centric dense prediction tasks (Figure 2).

**Encoder and Decoder.** We adopt the DINO series (Oquab et al., 2023) as the encoder $\mathcal{E}_{\mathrm{DINO}}$, which extracts global representations from the input image $\mathbf{x} \in \mathbb{R}^{H \times W \times 3}$ as $\mathbf{F}_{\mathrm{enc}} = \mathcal{E}_{\mathrm{DINO}}(\mathbf{x})$. The DPT (Ranftl et al., 2021) decoder $\mathcal{D}_{\mathrm{DPT}}$ transforms this representation into multi-scale features $\mathbf{F}_{\mathrm{DPT}} = \mathcal{D}_{\mathrm{DPT}}(\mathbf{F}_{\mathrm{enc}})$. On top of these features, we leverage three lightweight task heads $\mathcal{H}$ that produce predictions for depth, surface normals, and foreground/background segmentation $\{\hat{\mathbf{D}}, \hat{\mathbf{N}}, \hat{\mathbf{S}}\} = \mathcal{H}(\mathbf{F}_{\mathrm{DPT}})$. To capture the temporal relationship between frames, we inject four temporal blocks $\mathcal{T}$ into the decoder as bridges connecting different frames. The structure of the temporal blocks in the model is similar to that in AnimateDiff (Guo et al., 2023) and VDA (Chen et al., 2025), consisting of several temporal attention blocks.

**Local Geometry Enhancement.** While DINO tokens effectively encode semantic information and capture long-range dependencies, they generally lack fine details such as edges and textures. Inspired by the Resizer module in DAViD (Saleh et al., 2025), we introduce a lightweight CNN branch $\mathcal{E}_{\mathrm{CNN}}$. This branch directly extracts edges and textures from the input image as $\mathbf{F}_{\mathrm{CNN}} = \mathcal{E}_{\mathrm{CNN}}(\mathbf{x})$. The final fused representation is then obtained by concatenating the decoder features with the CNN features, followed by a nonlinear mapping: $\mathbf{F}_{\mathrm{fusion}} = \phi_{\mathrm{fusion}}([\mathbf{F}_{\mathrm{DPT}}, \mathbf{F}_{\mathrm{CNN}}])$, where $[\cdot, \cdot]$ denotes channel concatenation and $\phi$ denotes a nonlinear mapping.

**Channel Weight Adaptation (CWA).** While the fusion design preserves global semantics and strengthens texture cues, the lightweight CNN branch can introduce redundant appearance signals. DAViD observed similar issues, with appearance details such as tattoos and lighting patterns sometimes being mistaken for geometric shapes. To alleviate this, we introduce a channel weight adaptation module to reweight the channel weights of the fused features. Specifically, given the fused feature map $\mathbf{F}_{\mathrm{fusion}} \in \mathbb{R}^{C \times H \times W}$, we introduce a light-weight channel-wise reweighting block to adjust the contribution of each channel. We first apply global average pooling over the spatial dimensions to obtain a channel descriptor

$$q_c = \frac{1}{HW} \sum_{h=1}^{H} \sum_{w=1}^{W} \mathbf{F}_{\mathrm{fusion}}(c, h, w), \quad c = 1, \ldots, C, \tag{1}$$

which forms a vector $q \in \mathbb{R}^C$. This vector is then passed through a small two-layer MLP with a non-linear activation and a sigmoid function $\sigma(\cdot)$ to produce per-channel weights

$$a = \sigma(\mathrm{MLP}(q)) \in (0, 1)^C. \tag{2}$$

Figure 2: **Pipeline overview.** Given a sequence of RGB frames, our model extracts DINO features, global image features, and human geometric priors. These features are fused and re-weighted to generate enhanced representations for predicting temporally consistent depth, surface normals, and segmentation masks.

Finally, the fused features are rescaled channel-wise as

$$\mathbf{F}'_{\text{fusion}}(c, h, w) = a_c \, \mathbf{F}_{\text{fusion}}(c, h, w), \tag{3}$$

where $a_c$ denotes the weight of channel $c$. The CWA is trained jointly with depth and normal objectives, guiding the network to assign larger weights to channels. In this way, it reduces the weights of texture- and lighting-dominated channels while increasing the weights of geometry-related channels, thereby weakening the influence of appearance information on geometry prediction and maintaining the consistency of global representation.

**Human Geometric Prior.** Previous approaches mainly rely on general designs and data-centric scaling (larger and cleaner datasets), which raises the capacity from the data side but leaves model-side priors underused. We therefore inject a human-specific prior to strengthen the representation of the human body structure. A straightforward option is to use DensePose-like UV maps (Güler et al., 2018) so that the network predicts geometry for different body parts. However, due to the lack of such supervised data and in the multi-task setting, this option usually fails to achieve stable convergence. Instead, we adopt CSE (Neverova et al., 2020) as a stable geometric prior. Given a human image, the CSE encoder $\mathcal{E}_{\text{CSE}}$ produces continuous geometric embeddings $\mathbf{z} = \mathcal{E}_{\text{CSE}}(\mathbf{x})$, which we fuse with decoder features to impose shape-aware constraints on the predictions. Let $F_{\text{DPT}}$ denote the decoder features. To inject the human geometric prior into the representation, we project $\mathbf{z}$ to the same channel dimension and spatial resolution as $F_{\text{DPT}}$ using a $1 \times 1$ convolution followed by bilinear upsampling, and then fuse it with the decoder features by element-wise addition:

$$\tilde{\mathbf{z}} = \psi(\mathbf{z}) \in \mathbb{R}^{C \times H \times W}. \tag{4}$$

The prior is then fused with the decoder features by element-wise addition:

$$\mathbf{F}'_{\text{DPT}} = \mathbf{F}_{\text{DPT}} + \tilde{\mathbf{z}}. \tag{5}$$

## 3.3 TRAINING PIPELINE

To achieve multi-task human-centered temporal consistency, we adopt a two-stage training strategy. In stage 1, the model is pretrained on synthetic image data to learn spatially consistent fundamental representations. In stage 2, we inject the temporal module and continue training on synthetic video data with flow-guided stabilization term to capture temporal information and maintain consistency.

### 3.3.1 STAGE 1: STATIC IMAGE MODEL TRAINING

**Monocular Depth Estimation.** For depth estimation, given a depth map $\mathbf{d}^*$, we normalize it to the range $[0, 1]$ by $\mathbf{d} = \frac{\mathbf{d}^* - \min(\mathbf{d}^*)}{\max(\mathbf{d}^*) - \min(\mathbf{d}^*)}$. Let $\hat{\mathbf{D}}$ be the predicted relative depth. We follow previous work (Birkl et al., 2023) to estimate per-image scale and shift $(s, t)$. The depth loss is:

$$\mathcal{L}_{\text{depth}} = \left\| s\,\hat{\mathbf{D}} + t - \mathbf{d} \right\|_2 + \omega_{\text{grad}} \, \mathcal{L}_{\text{grad}}(s\hat{\mathbf{D}} + t, \, \mathbf{d}), \tag{6}$$

where $\mathcal{L}_{\text{grad}}$ is the gradient term (Hu et al., 2019) to encourage sharp boundaries and local continuity.

**Surface Normal Estimation.** The normal head outputs 3-channels $(x, y, z)$. Let $\mathbf{N}$ be the ground-truth normal and $\hat{\mathbf{N}}$ the prediction. The base loss combines a $L_1$ term with a cosine term:

$$\mathcal{L}_{\text{base}} = \|\mathbf{N} - \hat{\mathbf{N}}\|_1 + \left(1 - \mathbf{N} \cdot \hat{\mathbf{N}}\right). \tag{7}$$

Table 1: **Quantitative comparison** for depth estimation on THuman2.1 and Hi4D dataset. Note that the parameter size of Sapiens-0.3B is equivalent to that of large models of ViT-based methods.

| Methods | TH2.1-Face | | TH2.1-UpperBody | | TH2.1-FullBody | | Hi4D | |
|---|---|---|---|---|---|---|---|---|
| | RMSE↓ | AbsRel↓ | RMSE↓ | AbsRel↓ | RMSE↓ | AbsRel↓ | RMSE↓ | AbsRel↓ |
| DA-B | 0.0267 | 0.0157 | 0.0324 | 0.0175 | 0.0366 | 0.0176 | 0.0954 | **0.0251** |
| DA2-B | 0.0328 | 0.0204 | 0.0423 | 0.0241 | 0.0404 | 0.0209 | 0.0930 | 0.0262 |
| MoGe2-B | 0.0274 | 0.0165 | 0.0326 | 0.0179 | 0.0451 | 0.0208 | 0.1104 | 0.0281 |
| DAViD-B | 0.0254 | 0.0147 | 0.0262 | 0.0143 | 0.0304 | 0.0148 | 0.0947 | 0.0266 |
| Ours-B | **0.0193** | **0.0112** | **0.0228** | **0.0126** | **0.0293** | **0.0146** | **0.0928** | 0.0277 |
| DA-L | 0.0236 | 0.0138 | 0.0297 | 0.0162 | 0.0323 | 0.0160 | 0.0845 | 0.0228 |
| DA2-L | 0.0303 | 0.0187 | 0.0381 | 0.0216 | 0.0379 | 0.0197 | 0.0844 | 0.0239 |
| MoGe-L | 0.0222 | 0.0132 | 0.0276 | 0.0145 | 0.0361 | 0.0159 | 0.0915 | 0.0216 |
| MoGe2-L | 0.0231 | 0.0136 | 0.0294 | 0.0154 | 0.0349 | 0.0149 | 0.0892 | **0.0208** |
| DAViD-L | 0.0256 | 0.0149 | 0.0262 | 0.0144 | 0.0293 | 0.0142 | 0.0889 | 0.0244 |
| Sapiens-0.3B | 0.0150 | 0.0089 | 0.0184 | 0.0105 | 0.0239 | 0.0117 | 0.1349 | 0.0412 |
| Ours-L | **0.0147** | **0.0086** | **0.0174** | **0.0098** | **0.0218** | **0.0110** | **0.0700** | 0.0208 |
| Sapiens-0.6B | 0.0152 | 0.0087 | 0.0183 | 0.0104 | 0.0236 | 0.0119 | 0.1317 | 0.0407 |
| Sapiens-1B | 0.0119 | 0.0067 | 0.0145 | 0.0080 | 0.0179 | 0.0087 | 0.1151 | 0.0356 |
| Sapiens-2B | 0.0112 | 0.0061 | 0.0156 | 0.0086 | 0.0172 | 0.0082 | 0.1060 | 0.0327 |

We observe that when depth and normal heads are trained jointly, the predicted normals often lose fine texture details. This occurs because depth supervision relies on global geometric consistency and largely ignores high-frequency signals that are uninformative for relative depth. Since depth typically converges faster and more stably than normal estimation, it tends to dominate the shared representation during training. Consequently, the learned features emphasize smooth, low-frequency structures while suppressing texture cues, leading to over-smoothed surface normals, particularly on clothing, accessories, and hair. To mitigate this effect and enhance spatial coherence, we introduce an edge-aware gradient loss and a multi-scale Laplacian loss. Let $\nabla$ denote the Sobel operator and $\Delta$ the discrete Laplacian. Define an edge weight using the magnitude of ground-truth normal gradients $w_{\text{edge}} = 1 + \eta \frac{\|\nabla \mathbf{N}\| - \min\|\nabla \mathbf{N}\|}{\max\|\nabla \mathbf{N}\| - \min\|\nabla \mathbf{N}\|}$. The regularizers are:

$$\mathcal{L}_{\text{grad}}^{n} = w_{\text{edge}} \left\| \nabla(\hat{\mathbf{N}}) - \nabla(\mathbf{N})(x) \right\|_1 \quad \mathcal{L}_{\text{lap}} = w_{\text{edge}} \left\| \Delta(\hat{\mathbf{N}}) - \Delta(\mathbf{N}) \right\|_1. \tag{8}$$

The surface normal loss is:

$$\mathcal{L}_{\text{normal}} = \mathcal{L}_{\text{base}} + \alpha \, \mathcal{L}_{\text{grad}}^{n} + \beta \mathcal{L}_{\text{lap}}, \tag{9}$$

where $\alpha$ and $\beta$ are regularizers weights.

**Foreground Segmentation.** To provide human-centric foreground guidance for geometry-related tasks, we introduce a lightweight segmentation head that predicts a soft mask $\hat{\mathbf{S}}$ over the human region. Designed as an auxiliary branch that shares the same backbone as the depth and normal heads, this head supplies soft human masks that guide the depth and normal predictors to focus on the foreground and obtain cleaner supervision near human boundaries. The segmentation head predicts a soft mask $\hat{\mathbf{S}}$. We use binary cross-entropy to supervise them:

$$\mathcal{L}_{\text{seg}} = \mathcal{L}_{\text{BCE}}(\hat{\mathbf{S}}, \mathbf{S}), \tag{10}$$

and use $\mathbf{S}$ as the mask for depth and surface normal supervision. Finally, the Stage-1 objective is:

$$\mathcal{L}_{\text{stage1}} = \lambda_d \mathcal{L}_{\text{depth}} + \lambda_n \mathcal{L}_{\text{normal}} + \lambda_s \mathcal{L}_{\text{seg}}. \tag{11}$$

### 3.3.2 STAGE 2: DYNAMIC VIDEO MODEL TRAINING

To address frame-to-frame instability in dense video prediction, existing methods can be broadly divided into two categories. The first category is the TGM loss proposed in VDA (Chen et al., 2025), which enforces temporal consistency by constraining the depth gradient between adjacent

Ours-L DAViD-L Sapiens-2B

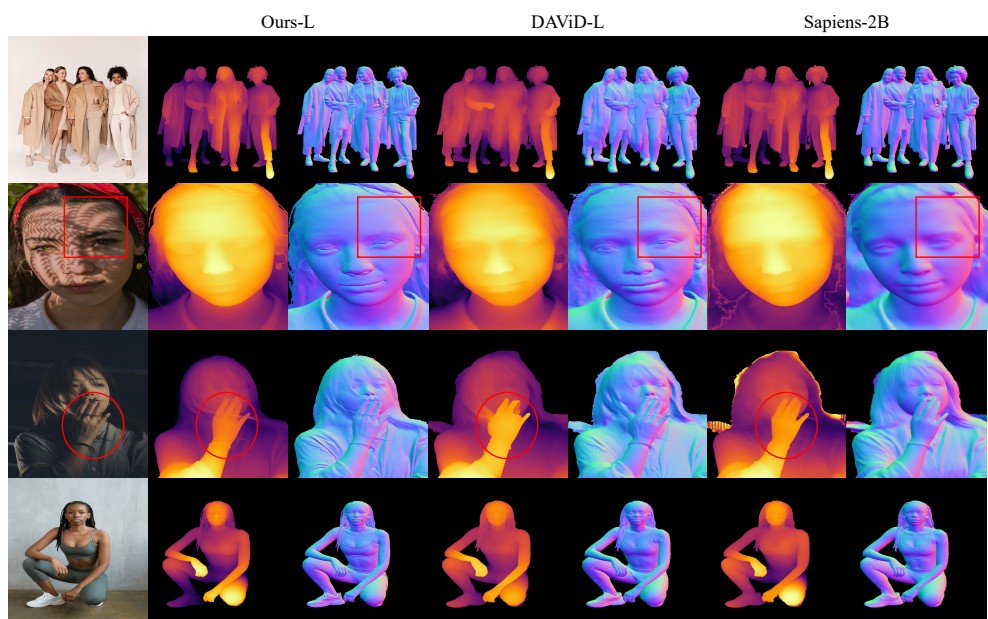

Figure 3: **Qualitative comparison** on challenging images in the wild.

frames. The second category is flow-based temporal consistency. Since our work involves not only depth but also normal estimation, with supervision primarily focused on human foreground regions, TGM is not directly suitable. It is restricted to depth prediction and tends to weaken supervision on fast-moving or occluded foreground regions. In contrast, flow-based methods explicitly establish correspondences across frames, which enables stable supervision in the foreground and naturally extends to enforcing directional consistency for surface normals.

Based on this, we keep all spatial losses and introduce optical-flow-based stabilization. For adjacent frames $k$ and $k+1$, we denote the forward and backward flows as $\mathcal{O}_{k\to k+1}$ and $\mathcal{O}_{k+1\to k}$. Warping with flow $\mathcal{O}$ is denoted as $\mathcal{W}(\cdot, \mathcal{O})$. To ensure reliable correspondences, we further apply a cycle-consistency mask $\mathcal{M}_{\mathrm{cyc}} = \mathbf{1}\Big(\|\mathcal{O}_{k\to k+1}(\mathcal{O}_{k+1\to k}) - \mathbf{x}\|_2 \le \tau_c\Big)$. We also suppress unstable boundary pixels using a non-edge mask from predicted depth edges. Let $\mathbf{E}_k$ be the edge map extracted from the current predicted depth, and let its dilated form be used to compute the edge mask $\mathcal{M}_{\mathrm{edge}} = 1 - \mathrm{Dilate}(\mathbf{E}_k)$. The valid set is $\mathcal{M} = \mathcal{M}_{\mathrm{cyc}} \cap \mathcal{M}_{\mathrm{edge}}$. Depth stabilization uses a bidirectional, flow-aligned $L_1$ loss:

$$
\begin{aligned}
\mathcal{L}_{\mathrm{temp}}^d = \frac{1}{|\mathcal{M}|} \left\| \mathcal{M} \odot \big( \hat{\mathbf{D}}_k - \mathcal{W}(\hat{\mathbf{D}}_{k+1}, \mathcal{O}_{k\to k+1}) \big) \right\|_1 \\
+ \frac{1}{|\mathcal{M}|} \left\| \mathcal{M} \odot \big( \hat{\mathbf{D}}_{k+1} - \mathcal{W}(\hat{\mathbf{D}}_k, \mathcal{O}_{k+1\to k}) \big) \right\|_1,
\end{aligned}
\tag{12}
$$

which reduces flicker and drift in where corresponding. Similarly, surface normal stabilization term:

$$
\begin{aligned}
\mathcal{L}_{\mathrm{temp}}^n = \frac{1}{|\mathcal{M}|} \mathcal{M} \odot \big( 1 - \cos\langle \mathcal{W}(\hat{\mathbf{N}}_k, \mathcal{O}_{k\to k+1}), \ \hat{\mathbf{N}}_{k+1} \rangle \big) \\
+ \frac{1}{|\mathcal{M}|} \mathcal{M} \odot \big( 1 - \cos\langle \mathcal{W}(\hat{\mathbf{N}}_{k+1}, \mathcal{O}_{k+1\to k}), \ \hat{\mathbf{N}}_k \rangle \big).
\end{aligned}
\tag{13}
$$

This term uses a smaller weight than the depth temporal term to suppress random directional jitter without oversmoothing true edges. Finally, the Stage-2 objective is:

$$
\mathcal{L}_{\mathrm{stage2}} = \mathcal{L}_{\mathrm{stage1}} + \lambda_{\mathrm{temp}}^d \mathcal{L}_{\mathrm{temp}}^d + \lambda_{\mathrm{temp}}^n \mathcal{L}_{\mathrm{temp}}^n.
\tag{14}
$$

Table 2: **Quantitative comparison** for surface normal estimation on THuman2.1 and Hi4D dataset. Note that the parameter size of Sapiens-0.3B is equivalent to that of large models of ViT-based methods.

| Methods | THuman2.1 | | | | | Hi4D | | | | |
| | Angular Error (°) ↓ | | % Within $t°$ ↑ | | | Angular Error (°) ↓ | | % Within $t°$ ↑ | | |
| | Mean | Median | 11.25° | 22.5° | 30° | Mean | Median | 11.25° | 22.5° | 30° |
|---|---|---|---|---|---|---|---|---|---|---|
| MoGe2-B | 20.31 | 17.94 | 27.04 | 64.96 | 81.30 | 19.29 | 15.52 | 33.52 | 72.03 | 85.31 |
| DAViD-B | 19.85 | 16.89 | 31.38 | 67.40 | 81.56 | 20.64 | 16.10 | 32.14 | 69.69 | 82.70 |
| Ours-B | **17.89** | **15.56** | **32.98** | **73.69** | **87.15** | **16.08** | **12.03** | **47.76** | **81.49** | **89.98** |
| MoGe2-L | 18.21 | 16.00 | 31.95 | 72.01 | 86.41 | 17.26 | 13.60 | 40.40 | 78.61 | 88.92 |
| DAViD-L | 19.59 | 16.64 | 30.02 | 68.18 | 82.09 | 20.74 | 16.11 | 31.94 | 69.42 | 82.55 |
| Sapiens-0.3B | **14.34** | **11.84** | **49.60** | **83.79** | **92.07** | 20.01 | 15.42 | 34.41 | 71.58 | 83.90 |
| Ours-L | 16.00 | 13.51 | 41.00 | 79.79 | 90.04 | **15.00** | **10.84** | **53.56** | **84.27** | **91.15** |
| Sapiens-0.6B | 14.34 | 11.92 | 49.19 | 83.82 | 92.22 | 17.87 | 13.50 | 41.43 | 77.79 | 87.79 |
| Sapiens-1B | 13.36 | 10.91 | 54.06 | 86.31 | 93.38 | 15.50 | 10.96 | 52.93 | 83.74 | 90.66 |
| Sapiens-2B | 13.13 | 10.66 | 55.38 | 86.81 | 93.57 | 15.58 | 11.05 | 52.47 | 84.02 | 90.79 |

# 4 EXPERIMENTS

## 4.1 IMPLEMENTATION DETAILS

As described in Section 3.3, we train our model using both static and dynamic data. The static data consists of 2M samples from our synthetic dataset and 300K samples from the SynthHuman dataset (Saleh et al., 2025), while the dynamic data uses 4M samples from our synthetic dataset. We adopt the latest DINOv3 (Siméoni et al., 2025) as the pretrained weights. For the static image model, we train a ViT-L with a batch size of 128 for 50K steps, which takes about 2.5 days. For the dynamic video model, we use a batch size of 8 with 32 frames and train for 35K steps, which requires about 1.5 days. The detailed hyperparameters for both training stages are provided in the Section A.6.

## 4.2 EVALUATION PROTOCOL

**Evaluation Datasets.** We evaluate our method on two challenging real-world datasets, THuman2.1 (Yu et al., 2021) and Hi4D (Yin et al., 2023b), for validating depth estimation and surface normal estimation. Following the evaluation protocol in Sapiens (Khirodkar et al., 2024), we construct three subsets on THuman2.1, including face, upper-body, and full-body. Unlike prior works that mainly relied on THuman2.0 with only 500 models and 1,500 images, we adopt the latest THuman2.1 dataset, which contains 2,445 models. Based on these models, we synthesize 7,335 images, resulting in a dataset with a significantly larger scale. For Hi4D, we select sequences from subjects 28, 32, and 37 captured by camera 4, covering 6 different subjects and yielding 1,195 multi-person real images. For image evaluation, we employ both

Table 3: **Qualitative comparison** for video depth and surface normal estimation on Hi4D.

| Methods | Depth | | Normal | | |
| | OPW↓ | TC-RMSE↓ | OPW↓ | TC-Mean↓ | TC-Abs↓ |
|---|---|---|---|---|---|
| MoGe2-B | 0.0176 | 0.0283 | 0.0362 | 4.26 | 0.162 |
| MoGe2-L | 0.0176 | 0.0288 | 0.0363 | 4.27 | 0.146 |
| DAViD-B | 0.0176 | 0.0283 | 0.0423 | 4.92 | 0.170 |
| DAViD-L | 0.0176 | 0.0288 | 0.0423 | 4.93 | 0.170 |
| Sapiens-0.3B | 0.0145 | 0.0226 | 0.0594 | 6.91 | 0.164 |
| Sapiens-0.6B | 0.0165 | 0.0266 | 0.0486 | 5.64 | 0.147 |
| Sapiens-1B | 0.0141 | 0.0240 | 0.0452 | 5.26 | 0.147 |
| Sapiens-2B | 0.0122 | 0.0221 | 0.0421 | 4.89 | 0.149 |
| NormalCrafter | - | - | 0.0277 | 3.20 | 0.143 |
| DepthCrafter | 0.0111 | 0.0304 | - | - | - |
| VDA-B | 0.0111 | 0.0304 | - | - | - |
| VDA-L | 0.0102 | 0.0300 | - | - | - |
| Ours-B | 0.0072 | 0.0189 | 0.0280 | 3.27 | 0.140 |
| Ours-L | **0.0070** | **0.0166** | **0.0261** | **3.04** | **0.133** |

THuman2.1 and Hi4D to assess depth and surface normal estimation under static poses. For video evaluation, we utilize Hi4D, which also provides temporally continuous dynamic sequences, enabling us to further examine the adaptability and generalization of our method in dynamic scenarios.

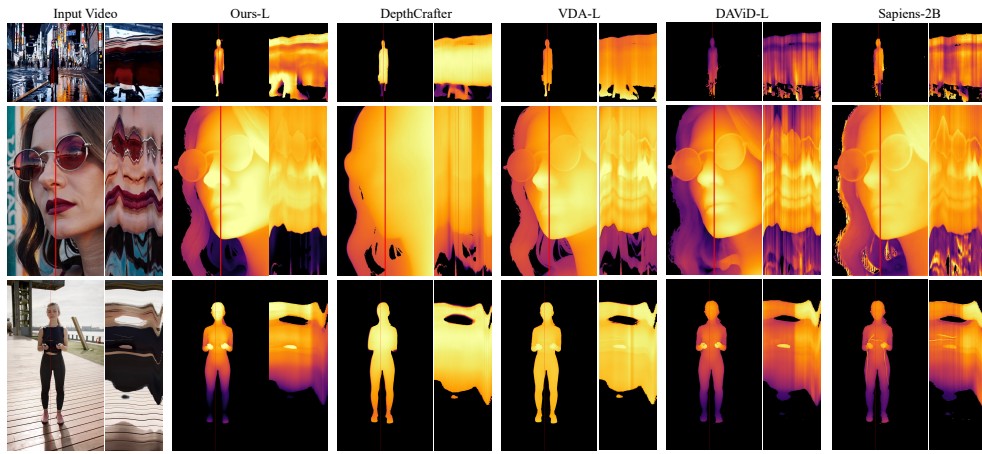

Figure 4: **Qualitative comparison** on video depth estimation. For better visualization, we also show the time slice on the red lines of each video on their right side.

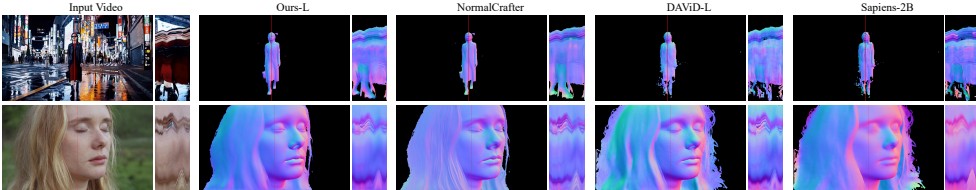

Figure 5: **Qualitative comparison** on video surface normal estimation. For better visualization, we also show the time slice on the red lines of each video on their right side.

**Evaluation Metric.** Following previous work (Khirodkar et al., 2024), to evaluate image depth estimation, we report the mean absolute value of the relative depth (AbsRel) and the root mean square error (RMSE). To evaluate image surface normal estimation, we use the standard metrics of mean and median angular error, as well as the percentage of pixels within $t^\circ$ error for $t \in \{11.25, 22.5, 30\}$. For video depth and surface normal estimation, we further consider temporal consistency across frames. We employ optical flow-based metrics computed using RAFT (Teed & Deng, 2020). We report the optical flow-based warping metric (OPW) (Wang et al., 2022), which measures the discrepancy between consecutive frames after warping. For depth frames, we report the flow-based temporal consistency error (TC-RMSE), which measures the stability of depth predictions across time. For normal frames, we report the flow-based angular error (TC-Mean), which evaluates the temporal consistency of surface normals. However, it should be noted that TC-Mean may be inaccurate if the predicted surface normals are globally biased or too smooth, especially when only evaluating the foreground. Thus, we introduce a new temporal consistency metric for surface normals. Based on the flow-based angular error, we compute the ground truth angular error and compare it with the predicted angular error using the absolute difference (TC-Abs). This metric reflects the discrepancy between predicted and ground truth temporal changes in surface orientation. Unlike purely flow-warped metrics, it can partly mitigate the influence of flow inaccuracies and place more emphasis on whether the temporal variations in predictions follow the ground truth.

### 4.3 COMPARISON TO THE STATE-OF-THE-ART

For static depth estimation, we evaluate several SOTA models, including general-purpose approaches (the Depth Anything family (Yang et al., 2024a;b) and the Moge family (Wang et al., 2025a;b)) as well as human-centric methods (Sapiens Khirodkar et al. (2024) and DAViD (Saleh et al., 2025)). As shown in Table 1, both variants of our model outperform these baselines on both datasets. Notably, our Large model achieves comparable or even superior accuracy on Hi4D static depth compared to the larger Sapiens-0.6B/1B/2B, highlighting its parameter efficiency and strong cross-dataset generalization. For static surface normal estimation, the results in Table 2 show that Sapiens performs particularly well on the THuman2.1 dataset, likely due to its similarity to RenderPeople, which was used during fine-tuning. On the Hi4D dataset, however, our Large model even surpasses Sapiens-2B. For soft foreground segmentation, we follow the experimental setting of DAViD to compare Zhong et al. (Zhong & Zharkov, 2024), BGMv2 (Lin et al., 2020), P3M-Net

Table 4: Comparison on the P3M-500-NP, P3M-500-P and PPM-100 benchmarks.

| Method | P3M-500-NP | | | P3M-500-P | | | PPM-100 | |
|---|---|---|---|---|---|---|---|---|
| | SAD ↓ | SAD-T ↓ | Conn ↓ | SAD ↓ | SAD-T ↓ | Conn ↓ | SAD ↓ | Conn ↓ |
| Zhong et al. | **10.60** | **6.83** | **9.77** | 10.04 | **6.44** | **9.41** | 90.28 | 84.09 |
| BGMv2 | 15.66 | 7.72 | 14.65 | 13.90 | 7.23 | 13.13 | 159.44 | 149.79 |
| P3M-Net | 11.23 | 7.65 | 12.51 | **8.73** | 6.89 | 13.88 | 142.74 | 139.89 |
| MODNet | 20.20 | 12.48 | 18.41 | 30.08 | 12.22 | 28.61 | 104.35 | 96.45 |
| DAViD | 14.83 | 10.23 | 14.76 | 12.65 | 9.19 | 12.47 | 78.17 | 74.72 |
| Ours | 13.12 | 11.88 | 12.72 | 11.63 | 9.95 | 11.51 | **70.71** | **68.32** |

(Ma et al., 2023), and MODNet (Ke et al., 2022), and we quantitatively evaluate our segmentation head on the P3M-500-NP, P3M-500-P, and PPM-100 benchmark datasets. As shown in Table 4, our method shows competitive results on both P3M validation sets and gives clear gains over DAViD and the other baselines on PPM-100, where our approach decreases SAD from 78.17 to 70.71 and Conn from 74.72 to 68.32. Since there is currently no released video human-centric model for depth or surface normal estimation, we compare against SOTA models designed for general scene videos, such as NormalCrafter (Bin et al., 2025), DepthCrafter (Hu et al., 2025), and VDA (Chen et al., 2025). As shown in Table 3, our model demonstrates superior performance in scenes containing humans. Figure 3 demonstrates the robustness of our method when tested on person-centric images, covering normal images, shadows, lighting changes, and multi-person scenes. Figure 4 and Figure 5 demonstrate the excellent performance of our method on human-centric in-the-wild Internet videos and the temporal consistency, respectively.

### 4.4 Ablation Studies

We mainly conduct the ablation studies on the Hi4D dataset. As shown in Table 5, we compare our full model against three variants: A) DPT head with an additional CNN branch as the baseline; B) w/ Human CSE prior; C) w/ channel weight adaptation. The results indicate that incorporating human structural priors through CSE

Table 5: **Ablation** on Hi4D dataset.

| Methods | Depth | | Normal | | | | |
|---|---|---|---|---|---|---|---|
| | RMSE ↓ | AbsRel ↓ | Mean ↓ | Median ↓ | 11.25° ↑ | 22.5° ↑ | 30° ↑ |
| Baseline | 0.0964 | 0.0279 | 20.51 | 16.00 | 32.22 | 70.12 | 82.74 |
| w/ CSE | 0.0932 | 0.0274 | 17.97 | 14.33 | 40.57 | 76.98 | 88.00 |
| w/ CWA | 0.0944 | 0.0271 | 18.32 | 15.82 | 42.31 | 77.43 | 88.56 |
| Full | 0.0928 | 0.0277 | 16.08 | 12.03 | 47.76 | 81.49 | 89.98 |

encourages the model to capture geometry that aligns with human body shape and articulation, which strengthens local surface details and orientation consistency. On the other hand, CWA emphasizes adaptive feature reweighting across channels, which improves the prediction stability. More detail ablation please refer to Section A.4.

## 5 Conclusion

This work presented a framework for human-centric dense prediction with temporal consistency. By constructing a synthetic pipeline that produces static frames and dynamic sequences with pixel-accurate annotations, we enabled joint learning of segmentation, depth, and surface normals with both spatial accuracy and stable video performance. Our model achieves strong results on THuman2.1 and Hi4D, and generalizes to in-the-wild videos. The results indicate that large-scale synthetic data, together with temporal supervision and human priors, can be an effective approach for improving human-centric video perception. In future work, we plan to extend the framework to more complex scenes and examine its use in downstream tasks such as human 3D reconstruction.

## 6 Ethics statement

This research does not involve human subjects, personally identifiable information, or sensitive data. The datasets and models used are publicly available and commonly adopted in the research community. No experiments were conducted that could directly or indirectly cause harm to individuals,

groups, or the environment. We have taken care to ensure fairness, reproducibility, and compliance with ethical research standards.

## 7 REPRODUCIBILITY STATEMENT

We detail our data pipeline and model components in the Section 3 and the training parameters in the Section A.6. Furthermore, we will release the source code and pretrained model weights upon the paper's acceptance.

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

# A APPENDIX

## A.1 DATA SYNTHESIS PIPELINE

The full data synthesis pipeline is illustrated in Figure 6. The process begins with the construction of clothed human models using character-generation tools such as DAZ 3D, MakeHuman, and Character Creator. These tools provide parametric control over body shape and garment categories, forming the base set of assets used for large-scale identity sampling.

To expand the appearance diversity, we apply texture augmentations on the diffuse maps. These operations include color-based perturbations and material replacement, which allow the same geometry to support a wide range of surface styles. After texture augmentation, each identity is paired with motion data by retargeting AMASS skeletal trajectories. This step assigns realistic human motion while maintaining consistent rigging across different characters.

The animated models are then placed into Blender, where we define camera poses, focal lengths, and tracking behavior. Randomization in these settings increases viewpoint variety in both static and dynamic supervision. During rendering, Blender outputs synchronized RGB images, depth maps, surface normals, and segmentation masks.

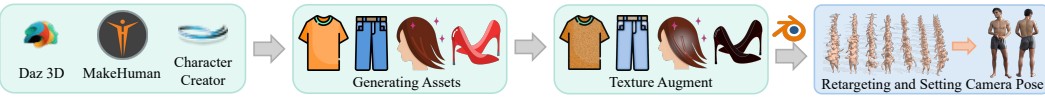

Figure 6: We first generate clothed human models using DAZ 3D, MakeHuman, and Character Creator. Texture augmentations are applied to increase appearance diversity. Each model is then animated by retargeting AMASS motion sequences. Finally, models are placed in Blender with randomized cameras for rendering RGB images together with depth, surface normals, and segmentation masks.

## A.2 DISCUSSION

Existing synthetic data pipelines for human-centric learning tasks primarily focus on static image generation or structural parameter supervision. SURREAL (Varol et al., 2017) combines SMPL models with MoCap sequences to produce synthetic videos with depth, surface normal, and part segmentation annotations. However, it lacks realistic clothing or hair geometry, relying on simplified texture mappings. PeopleSansPeople (Ebadi et al., 2021) leverages Unity to render large-scale, domain-randomized human images, supporting segmentation and keypoint labels but does not generate temporally aligned sequences or pixel-level geometric cues. SynBody (Yang et al., 2023) substantially improves the scale and diversity of identities and actions using SMPL-XL and layered clothing models, providing multi-view video sequences and mesh-level supervision. However, its released data focuses on RGB and pose annotations, with depth and normal modalities not included in the official release. More recent pipelines such as SynthMoCap (Hewitt et al., 2024) emphasize high-fidelity single-frame supervision for dense prediction tasks. They provide detailed annotations like depth, surface normals, and masks, but are limited to frame-level modeling without temporal continuity. Our pipeline is explicitly designed to generate temporally aligned, multi-modal human video sequences, enabling supervision for both per-frame and sequence-level tasks. We synthesize high-quality clothed human characters using commercial modeling tools with randomized sampling over body types, clothing, and textures to construct a large and diverse identity set. Motions from the AMASS dataset are retargeted to these characters and rendered in Blender to produce videos along with per-frame RGB, depth, surface normal, and segmentation maps. Unlike most prior work, our pipeline produces frame-consistent annotations, supporting dense, temporally stable supervision across tasks such as point tracking, normal prediction, and temporally coherent segmentation.

## A.3 ADDITIONAL QUALITATIVE RESULTS

In Figure 8 and Figure 9, we provide additional qualitative results. We also provide some video results in the supplementary materials.

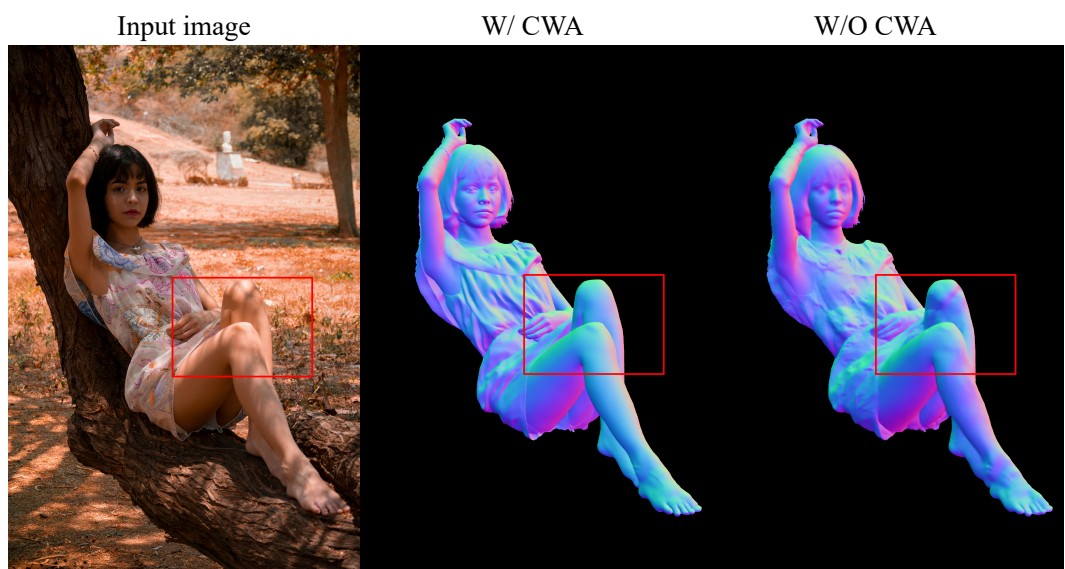

Figure 7: Ablation for Channel Weight Adaptation.

## A.4 ADDITIONAL ABLATION STUDY

**Investigate channel weight adaptation.** The motivation for introducing a CNN branch in the dual-branch structure is to compensate for the limited capability of the Transformer backbone in modeling local textures, similar to the design adopted in the DAViD (Saleh et al., 2025). CNNs are effective at capturing local patterns and edge details, which helps refine the quality of depth or normal predictions. In our experiments, we observed that adding the CNN branch improves surface continuity and local smoothness. However, this advantage also comes with a drawback, since CNNs tend to capture redundant texture signals that are not related to geometry, such as shadows, clothing patterns, or tattoos. When such signals are fused into the prediction, they interfere with the recovery of the underlying geometry and may produce artifacts or instability in challenging scenarios.

To address this issue, we introduce the CWA module into the CNN branch. CWA adaptively adjusts the channel-wise feature weights, suppressing those that contribute little or negatively to geometry recovery while emphasizing features that are strongly correlated with shape. In practice, CWA acts as a dynamic filter placed between the CNN branch and the final prediction, enabling the model to better distinguish between texture information and geometric cues. Comparative results in Figure 7 show that incorporating CWA effectively reduces artifacts in local regions, especially in cases with complex lighting or decorative textures, and leads to more stable and consistent predictions.

Table 6: Ablation for DINOv2 and DINOv3 on Thuman2.1 and Hi4D datasets using the depth task.

| Methods | TH2.1-Face | | TH2.1-UpperBody | | TH2.1-FullBody | | Hi4D | |
|---|---|---|---|---|---|---|---|---|
| | RMSE↓ | AbsRel↓ | RMSE↓ | AbsRel↓ | RMSE↓ | AbsRel↓ | RMSE↓ | AbsRel↓ |
| DINOv2-B | 0.0207 | 0.0105 | 0.0251 | 0.0116 | 0.0321 | 0.0136 | 0.0862 | 0.0228 |
| DINOv3-B | 0.0193 | 0.0108 | 0.0234 | 0.0116 | 0.0302 | 0.0135 | 0.0871 | 0.0212 |
| DINOv2-L | 0.0167 | 0.0088 | 0.0211 | 0.0102 | 0.0293 | 0.0123 | 0.0771 | 0.0193 |
| DINOv3-L | 0.0158 | 0.0085 | 0.0198 | 0.0098 | 0.0243 | 0.0111 | 0.0768 | 0.0195 |

**Investigate DINOv2 and DINOv3.** To ensure a fair comparison, we are using $518 \times 518$ input resolution for DINOv2 (Oquab et al., 2023) and $592 \times 592$ input resolution for DINOv3. As shown in Table 6, DINOv3 (Siméoni et al., 2025) achieves consistent improvements over DINOv2 across multiple benchmarks, particularly on Thuman-FullBody and Hi4D, where the results are more stable. While these gains partially benefit from the stronger encoder, it is important to highlight that the overall performance improvements of our approach do not stem solely from replacing the backbone.

Instead, they result from the joint contribution of our carefully designed model and the curated data used for training. This synergy allows the network to better handle the challenges of real-world scenarios, leading to more reliable geometry recovery and stronger generalization across datasets.

**Investigate human prior.** We investigate human priors on the depth task. As shown in Table Table 7, adding a Fourier UV map improves over the baseline, suggesting that canonical UV coordinates provide useful geometric cues for depth estimation. While our human prior achieves the more excellent results on both Thuman and Hi4D, reducing both RMSE and AbsRel. These results highlight that human-structure-aware priors enable more accurate and stable depth predictions compared to purely positional encodings.

Table 7: Ablation on human priors using the depth task with DINOv3-B.

| Methods | TH2.1 | | Hi4D | |
|---|---|---|---|---|
| | RMSE↓ | AbsRel↓ | RMSE↓ | AbsRel↓ |
| Baseline | 0.0223 | 0.0119 | 0.0964 | 0.0279 |
| + Fourier UV | 0.0192 | 0.0110 | 0.0922 | 0.0273 |
| + CSE | 0.0189 | 0.0108 | 0.0912 | 0.0271 |

**Investigate human prior fusion strategies.** We compare two strategies for integrating human priors, concatenation (cat) and addition (add). As shown in Table 8, both strategies improve depth and normal estimation, but addition consistently achieves better results. Specifically, add reduces both RMSE and AbsRel while also lowering the mean and median angular error for surface normals. The improvement can be attributed to the fact that addition enforces direct feature alignment between the prior and the learned representations, whereas concatenation requires the network to learn how to fuse heterogeneous features. This suggests that additive integration provides a more effective way to inject human-structure priors, yielding more stable and geometry-aware predictions.

Table 8: Ablation on different integration strategies of human priors on Hi4D with DINOv3-B.

| Methods | Depth | | Normal | | | | |
|---|---|---|---|---|---|---|---|
| | RMSE↓ | AbsRel↓ | Mean↓ | Median↓ | 11.25°↑ | 22.5°↑ | 30°↑ |
| Cat | 0.0955 | 0.0284 | 16.96 | 14.23 | 38.20 | 76.13 | 87.86 |
| Add | 0.0933 | 0.0280 | 16.41 | 13.66 | 40.83 | 77.57 | 88.67 |

**Investigate the impact of loss weight on multi-task.** We further investigate the effect of balancing depth and normal losses by varying the weights $\lambda_d$ and $\lambda_n$. As shown in Table 9, setting equal weights ($\lambda_d$=1, $\lambda_n$=1) gives the weakest performance, suggesting that treating the two tasks uniformly introduces conflicts in optimization. Reducing the normal weight to $\lambda_n$=0.5 keeps the depth metrics almost unchanged but leads to a noticeable drop in normal estimation accuracy, indicating that the depth signal dominates training. Increasing the normal weight ($\lambda_d$=0.5, $\lambda_n$=1) slightly improves surface normals compared to the 1:1 setting but does not yield significant gains. The best results are obtained when $\lambda_d$=1 and $\lambda_n$=0.1, where both depth and normal predictions improve. This demonstrates that depth supervision should remain the primary training signal, while a lightly weighted normal loss provides complementary regularization without overwhelming the optimization.

**Investigate the impact of training data size.** Table 10 reports the impact of training data size on Hi4D using ViT-B. We observe a consistent improvement across both depth and normal prediction as the number of training samples increases. For depth estimation, RMSE and AbsRel gradually decrease when scaling from 300K to 2M, showing that additional data helps the model capture finer geometric cues. A similar trend is observed in surface normal prediction, where both Mean and Median angular errors become smaller, while the percentage of pixels within 11.25°, 22.5°, and 30° steadily increases. These results suggest that enlarging the training set enhances generalization ability and reduces overfitting, even when the backbone is fixed. However, the gain becomes marginal when moving from 600K to 2M, indicating that data scaling alone may saturate and further improvements may require stronger architectures or better data diversity.

Table 9: Ablation for depth and normal loss weights on Hi4D with DINOv3-B.

| Methods | Depth | | Normal | | | | |
|---|---|---|---|---|---|---|---|
| | RMSE↓ | AbsRel↓ | Mean↓ | Median↓ | 11.25°↑ | 22.5°↑ | 30°↑ |
| $\lambda_d=1$, $\lambda_n=1$ | 0.0955 | 0.0284 | 16.96 | 14.23 | 38.20 | 76.13 | 87.86 |
| $\lambda_d=1$, $\lambda_n=0.5$ | 0.0932 | 0.0281 | 17.25 | 14.50 | 37.60 | 75.98 | 87.40 |
| $\lambda_d=0.5$, $\lambda_n=1$ | 0.0947 | 0.0283 | 16.70 | 14.05 | 38.85 | 76.80 | 87.95 |
| $\lambda_d=1$, $\lambda_n=0.1$ | 0.0929 | 0.0279 | 16.61 | 13.76 | 40.03 | 77.37 | 88.55 |

Table 10: Ablation for different data size on Hi4D with DINOv3-B.

| DataSize | Depth | | Normal | | | | |
|---|---|---|---|---|---|---|---|
| | RMSE↓ | AbsRel↓ | Mean↓ | Median↓ | 11.25°↑ | 22.5°↑ | 30°↑ |
| Our [300K] | 0.0963 | 0.0286 | 17.10 | 14.40 | 38.10 | 76.10 | 87.80 |
| Our [600K] | 0.0954 | 0.0284 | 16.90 | 14.20 | 38.80 | 76.60 | 88.05 |
| Our [2M] | 0.0943 | 0.0282 | 16.70 | 13.90 | 39.50 | 76.90 | 88.15 |
| SynthHuman | 0.0971 | 0.0287 | 17.25 | 14.55 | 37.80 | 75.90 | 87.40 |
| SynthHuman + Our [300K] | 0.0958 | 0.0285 | 17.00 | 14.32 | 38.45 | 76.30 | 87.95 |
| SynthHuman + Our [600K] | 0.0946 | 0.0283 | 16.72 | 14.05 | 39.30 | 76.95 | 88.25 |
| SynthHuman + Our [2M] | 0.0940 | 0.0281 | 16.58 | 13.70 | 40.10 | 77.20 | 88.30 |

**Investigate the impact of normal regularization term.** Table 11 evaluates the effect of introducing a normal regularization term (NRT) in training. While the depth estimation metrics remain nearly unchanged, we observe a significant improvement in surface normal prediction. This indicates that the NRT provides strong geometric guidance, making the network more sensitive to local surface orientation without sacrificing depth accuracy. The results highlight that explicit geometric priors can complement photometric supervision and lead to better normal recovery, even under the same backbone capacity.

Table 11: Ablation for normal regularization term on Hi4D with DINOv3-B.

| Methods | Depth | | Normal | | | | |
|---|---|---|---|---|---|---|---|
| | RMSE↓ | AbsRel↓ | Mean↓ | Median↓ | 11.25°↑ | 22.5°↑ | 30°↑ |
| w/o NRT | 0.0938 | 0.0282 | 16.85 | 14.12 | 38.10 | 76.05 | 87.70 |
| w/ NRT | 0.0935 | 0.0281 | 15.40 | 12.95 | 45.25 | 79.85 | 89.45 |

**Investigate the impact of the flow stable term.** Table 12 investigates the role of different temporal losses. Without temporal regularization, both depth and surface normal predictions are unstable, yielding results on par with Sapiens-1B. Introducing the GMT loss improves depth consistency, with OPW and TC-RMSE reduced compared to the variant without temporal loss. However, the overall performance remains worse than VDA-B, and the normal metrics show almost no improvement. This is because GMT originates from the TGM loss in VDA, which constrains only depth gradients across adjacent frames. Such a method is limited to depth prediction and weakens supervision on fast-moving or occluded human foreground regions, making it unsuitable for surface normal estimation. In contrast, our flow-based temporal loss explicitly establishes correspondences across frames, enabling stable supervision in the foreground and naturally extending to directional consistency for surface normals.

**Investigate the impact of the proposed module.** We further evaluate the effect of the proposed components on the Thuman 2.1 dataset with DINOv3-B. As shown in Table 13, adding Human CSE prior consistently improves both depth and normal estimation over the baseline, for example reducing RMSE from 0.0266 to 0.0231 and lowering the mean normal error from 20.21° to 19.45°, while also increasing the percentage of normals within small angular thresholds. The CWA module also brings clear gains, especially on normal metrics (e.g., 11.25° and 22.5°), indicating that adaptive channel reweighting helps the network produce more stable and accurate surface orientation. These

Table 12: Ablation for video depth and surface normal estimation with different losses on Hi4D dataset.

| Methods | Depth | | Normal | | |
|---|---|---|---|---|---|
| | OPW↓ | TC-RMSE↓ | OPW↓ | TC-Mean↓ | TC-Abs↓ |
| w/o $\mathcal{L}_{temp}$ | 0.0144 | 0.0242 | 0.0450 | 5.22 | 0.148 |
| w/ $\mathcal{L}_{GMT}$ | 0.0120 | 0.0310 | 0.0455 | 5.28 | 0.150 |
| w/ $\mathcal{L}_{temp}$ | 0.0075 | 0.0191 | 0.0286 | 3.30 | 0.140 |

results confirm that the proposed modules generalize well beyond Hi4D and remain effective on Thuman 2.1.

Table 13: Ablation on Thuman 2.1 dataset with DINOv3-B.

| Methods | Depth | | Normal | | | | |
|---|---|---|---|---|---|---|---|
| | RMSE ↓ | AbsRel ↓ | Mean ↓ | Median ↓ | 11.25° ↑ | 22.5° ↑ | 30° ↑ |
| Baseline | 0.0266 | 0.0154 | 20.21 | 17.83 | 28.20 | 66.18 | 83.33 |
| w/ CSE | 0.0231 | 0.0134 | 19.45 | 16.52 | 30.10 | 69.72 | 85.02 |
| w/ CWA | 0.0248 | 0.0138 | 18.65 | 16.13 | 31.75 | 72.10 | 86.10 |
| Full | 0.0225 | 0.0122 | 17.89 | 15.56 | 32.98 | 73.69 | 87.15 |

**Investigate the impact of the temporal layer.** We conduct an ablation study to evaluate the contribution of the temporal layer (TL) to temporal consistency in depth and normal prediction. As shown in Table 14, removing the TL leads to clear degradation across all temporal metrics. In particular, TC-RMSE increases from 0.0189 to 0.0276 in depth, and TC-Mean rises from 3.27 to 4.55 in normal estimation. This confirms that the TL improves temporal stability by leveraging sequential information, especially for frames with fast motion or occlusion.

Table 14: Ablation on the effect of the temporal layer (TL) on Hi4D dataset with DINOv3-B.

| Methods | Depth | | Normal | | |
|---|---|---|---|---|---|
| | OPW↓ | TC-RMSE↓ | OPW↓ | TC-Mean↓ | TC-Abs↓ |
| w TL | 0.0072 | 0.0189 | 0.0280 | 3.27 | 0.140 |
| w/o TL | 0.0155 | 0.0276 | 0.0405 | 4.55 | 0.158 |

**Investigate the impact of the CNN branch.** We evaluate the effect of adding a dedicated CNN branch to complement the transformer backbone. As shown in Table 15, removing the CNN branch results in performance drops across both depth and normal estimation tasks. For depth, RMSE increases from 0.0964 to 0.0998 and AbsRel rises from 0.0279 to 0.0320. For normal prediction, the angular error metrics (Mean and Median) also degrade, and accuracy under angular thresholds (11.25°, 22.5°, 30°) drops consistently. These results confirm that the local inductive bias brought by the CNN branch helps refine fine-grained structures, especially around object boundaries, which complements the global modeling ability of the transformer backbone.

Table 15: Ablation for CNN branch on Hi4D dataset with DINOv3-B.

| Methods | Depth | | Normal | | | | |
|---|---|---|---|---|---|---|---|
| | RMSE ↓ | AbsRel ↓ | Mean ↓ | Median ↓ | 11.25° ↑ | 22.5° ↑ | 30° ↑ |
| w/ CNN | 0.0964 | 0.0279 | 20.51 | 16.00 | 32.22 | 70.12 | 82.74 |
| w/o CNN | 0.0998 | 0.0320 | 23.33 | 19.21 | 28.05 | 66.49 | 77.03 |

**Investigate the impact of the foreground segmentation branch.** We investigate how the auxiliary foreground segmentation head influences depth and normal prediction on Hi4D and THuman2.1 (Table 16). With the segmentation branch, depth RMSE / AbsRel decrease from 0.0963 / 0.0301 to 0.0928 / 0.0277 on Hi4D and from 0.0237 / 0.0130 to 0.0225 / 0.0122 on THuman2.1. For surface normals, the segmentation branch also reduces mean and median angular errors and increases the percentage of pixels within 11.25°, 22.5°, and 30° on both datasets. These consistent gains show

that the soft human masks predicted by this branch provide effective foreground guidance, allowing the geometry heads to focus on human regions and boundaries and thus improve overall geometry estimation quality.

Table 16: Ablation for foreground segmentation (FS) branch with DINOv3-B.

| Methods | Datasets | Depth | | Normal | | | | |
|---|---|---|---|---|---|---|---|---|
| | | RMSE ↓ | AbsRel ↓ | Mean ↓ | Median ↓ | 11.25° ↑ | 22.5° ↑ | 30° ↑ |
| w/o FS | Hi4D | 0.0963 | 0.0301 | 16.84 | 12.67 | 44.12 | 79.83 | 88.21 |
| w/ FS | Hi4D | 0.0928 | 0.0277 | 16.08 | 12.03 | 47.76 | 81.49 | 89.98 |
| w/o FS | Thuman 2.1 | 0.0237 | 0.0130 | 18.42 | 16.21 | 30.91 | 70.82 | 85.94 |
| w/ FS | Thuman 2.1 | 0.0225 | 0.0122 | 17.89 | 15.56 | 32.98 | 73.69 | 87.15 |

## A.5 MODEL PARAMETERS COMPARISON

We provide a comparison of model size and computational cost for representative state-of-the-art human-centric methods. Specifically, we report the number of parameters and GFLOPs for each model in Table 17, which highlights the relative complexity of our models compared with existing approaches.

Table 17: Model parameters comparison of SOTA human-centric methods.

| Methods | Params | GFLOPs |
|---|---|---|
| Sapiens-0.3B | 0.336 B | 1242 |
| Sapiens-0.6B | 0.664 B | 2583 |
| Sapiens-1B | 1.169 B | 4647 |
| Sapiens-2B | 2.163 B | 8709 |
| DAViD-B | 0.120 B | 344 |
| DAViD-L | 0.340 B | 663 |
| Ours-B | 0.097 B | 471 |
| Ours-L | 0.337 B | 753 |

## A.6 IMPLEMENTATION DETAILS

We use AdamW in both stages with weight decay 0.05 and $(\beta_1, \beta_2) = (0.9, 0.95)$. The learning rate schedule is a 2000-iteration linear warmup (start_factor $= 1/100$), followed by a polynomial decay (power $= 1.5$) until the end of training. More detail in Table 18 and Table 19.

## A.7 EVALUATION METRIC DETAILS

We employ standard metrics to quantitatively evaluate both static and video-based depth and surface normal estimation. Below we provide detailed definitions.

**Depth metrics.** Given predicted depth $\hat{D}$ and ground truth $D$, over valid pixels $M$, we adopt Root Mean Squared Error (RMSE) and Absolute Relative Error (AbsRel):

$$\text{RMSE} = \sqrt{\frac{1}{|M|} \sum_{i \in M} \left( \hat{D}_i - D_i \right)^2}, \tag{15}$$

$$\text{AbsRel} = \frac{1}{|M|} \sum_{i \in M} \frac{\left| \hat{D}_i - D_i \right|}{D_i}. \tag{16}$$

**Normal metrics.** Let $\hat{N}_i$ and $N_i$ denote predicted and ground truth normals at pixel $i$, both normalized to unit vectors. We report mean and median angular error, as well as threshold accuracies at

Table 18: Stage-1 (image) training hyperparameters.

| Hyperparameter | Value |
|---|---|
| Training step | 50,000 |
| Batch size | 128 |
| Learning rate (Encoder) | $1 \times 10^{-4}$ |
| Learning rate (Others) | $1 \times 10^{-5}$ |
| Weight decay | 0.05 |
| Optimizer | AdamW |
| Optimizer betas | (0.9, 0.95) |
| LR schedule | Linear + Polynomial |
| Freeze backbone | optional |
| Mask loss weight | 0.05 |
| Normal loss weight | 0.1 |

Table 19: Stage-2 (video) training hyperparameters.

| Hyperparameter | Value |
|---|---|
| Training step | 35,000 |
| Batch size (GPU) | 1 |
| Frames per clip | 32 |
| Learning rate (Temporal modules) | $1 \times 10^{-4}$ |
| Learning rate (Others) | $1 \times 10^{-6}$ |
| Weight decay | 0.05 |
| Optimizer | AdamW |
| Optimizer betas | (0.9, 0.95) |
| LR schedule | Linear + Polynomial |
| Freeze Encoder | True |
| Mask loss weight | 0.05 |
| Normal loss weight | 0.1 |
| Temporal Depth loss weight | 1 |
| Temporal Normal loss weight | 0.1 |

11.25°, 22.5°, and 30°:

$$\text{Mean} = \frac{1}{|M|} \sum_{i \in M} \arccos\left(\langle \hat{N}_i, N_i \rangle\right), \tag{17}$$

$$\text{Median} = \text{median}_{i \in M}\left(\arccos\left(\langle \hat{N}_i, N_i \rangle\right)\right), \tag{18}$$

$$\text{Acc}_\theta = \frac{1}{|M|} \sum_{i \in M} \mathbf{1}\left[\arccos\left(\langle \hat{N}_i, N_i \rangle\right) < \theta\right]. \tag{19}$$

**Video depth metrics.** For adjacent frames $k$ and $k+1$, with optical flow $O_{k \to k+1}$ and warping operator $W(\cdot, O)$, we define temporal consistency for depth as:

$$\text{OPW} = \frac{1}{|M|} \sum_{i \in M} \left| \hat{D}_k(i) - W(\hat{D}_{k+1}, O_{k \to k+1})(i) \right|, \tag{20}$$

$$\text{TC-RMSE} = \sqrt{\frac{1}{|M|} \sum_{i \in M} \left( \hat{D}_k(i) - W(\hat{D}_{k+1}, O_{k \to k+1})(i) \right)^2}. \tag{21}$$

**Video normal metrics.** Similarly, temporal consistency for normals is measured as:

$$\text{TC-Mean} = \frac{1}{|M|} \sum_{i \in M} \arccos \left( \langle \hat{N}_k(i), W(\hat{N}_{k+1}, O_{k \to k+1})(i) \rangle \right), \tag{22}$$

$$\text{TC-Abs} = \frac{1}{|M|} \sum_{i \in M} \Big| \arccos \big( \langle \hat{N}_k(i), W(\hat{N}_{k+1}, O_{k \to k+1})(i) \rangle \big) \tag{23}$$

$$- \arccos \big( \langle N_k(i), W(N_{k+1}, O_{k \to k+1})(i) \rangle \big) \Big|. \tag{24}$$

These metrics collectively measure accuracy at the frame level and temporal stability across frames for both depth and surface normal estimation.

## A.8 Limitations

Although the proposed approach improves overall stability and visual consistency, several limitations remain. The CNN branch, while effective at capturing local patterns, can also introduce redundant texture signals such as clothing patterns, decorative elements, or shadows. These signals interfere with the recovery of true geometry and may result in pseudo-geometric artifacts. The CWA module alleviates this issue by adaptively suppressing less informative channels and emphasizing features that are more relevant to geometry, but in scenes with highly complex textures, the effect is not completely eliminated. In addition, when large and rapid movements occur, such as turning, jumping, or swinging of limbs, occlusions and large displacements weaken temporal correspondences and lead to local instabilities in depth and normal predictions. A further difficulty arises in regions undergoing non-rigid deformations, including fluttering skirts, moving sleeves, or hair, where the complexity of local geometry and frequent occlusions still cause fluctuations and prediction biases.

## A.9 The Use of Large Language Models (LLMs)

Large language models (LLMs) were used only to assist with language editing and minor text improvements in the preparation of this manuscript. They were not involved in the design of the research, the development of methods, the execution of experiments, or the interpretation of results. All scientific content, including analyses, conclusions, and contributions, remains the work of the authors.

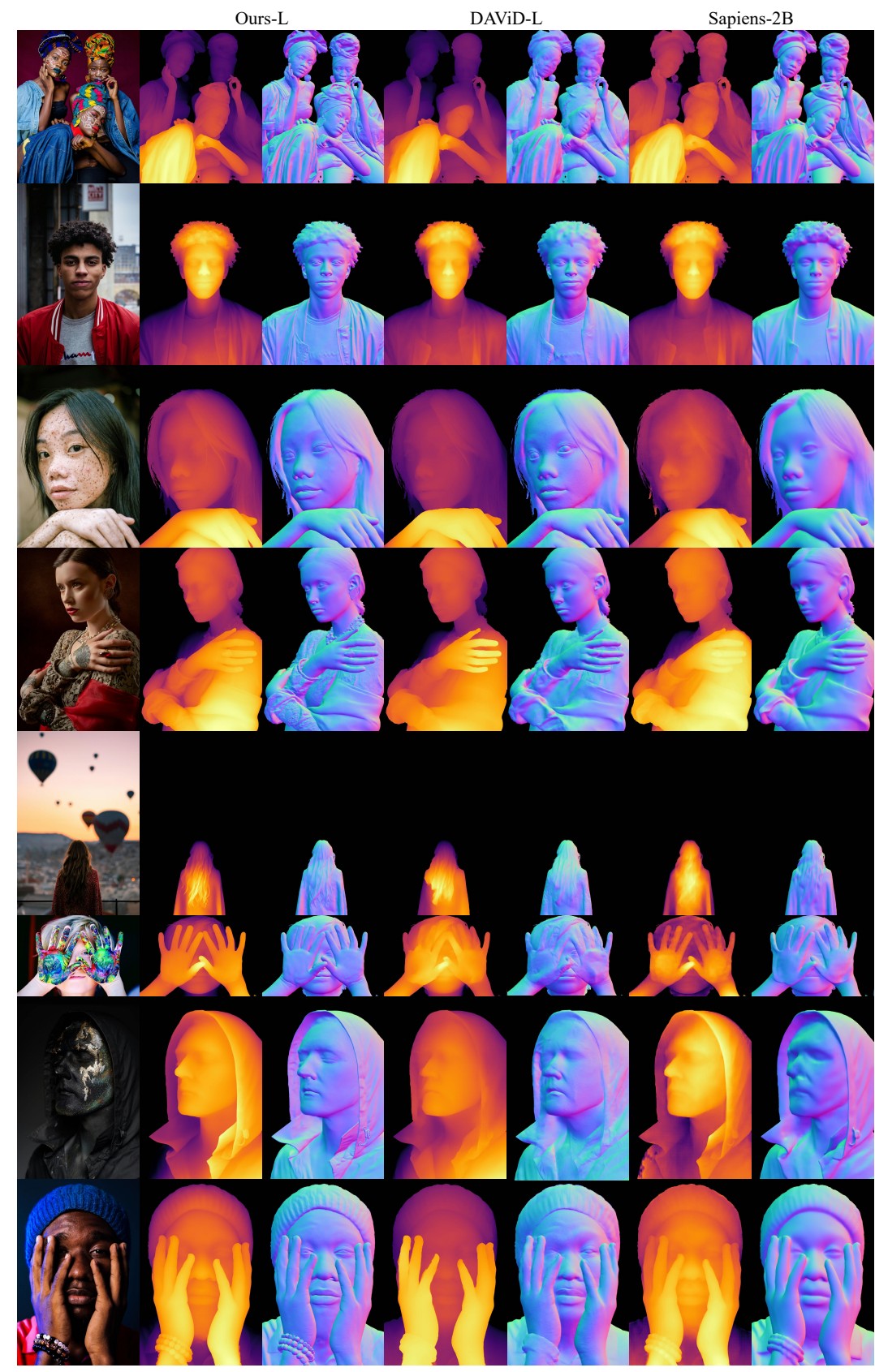

Figure 8: Additional qualitative comparison on challenging images in the wild.

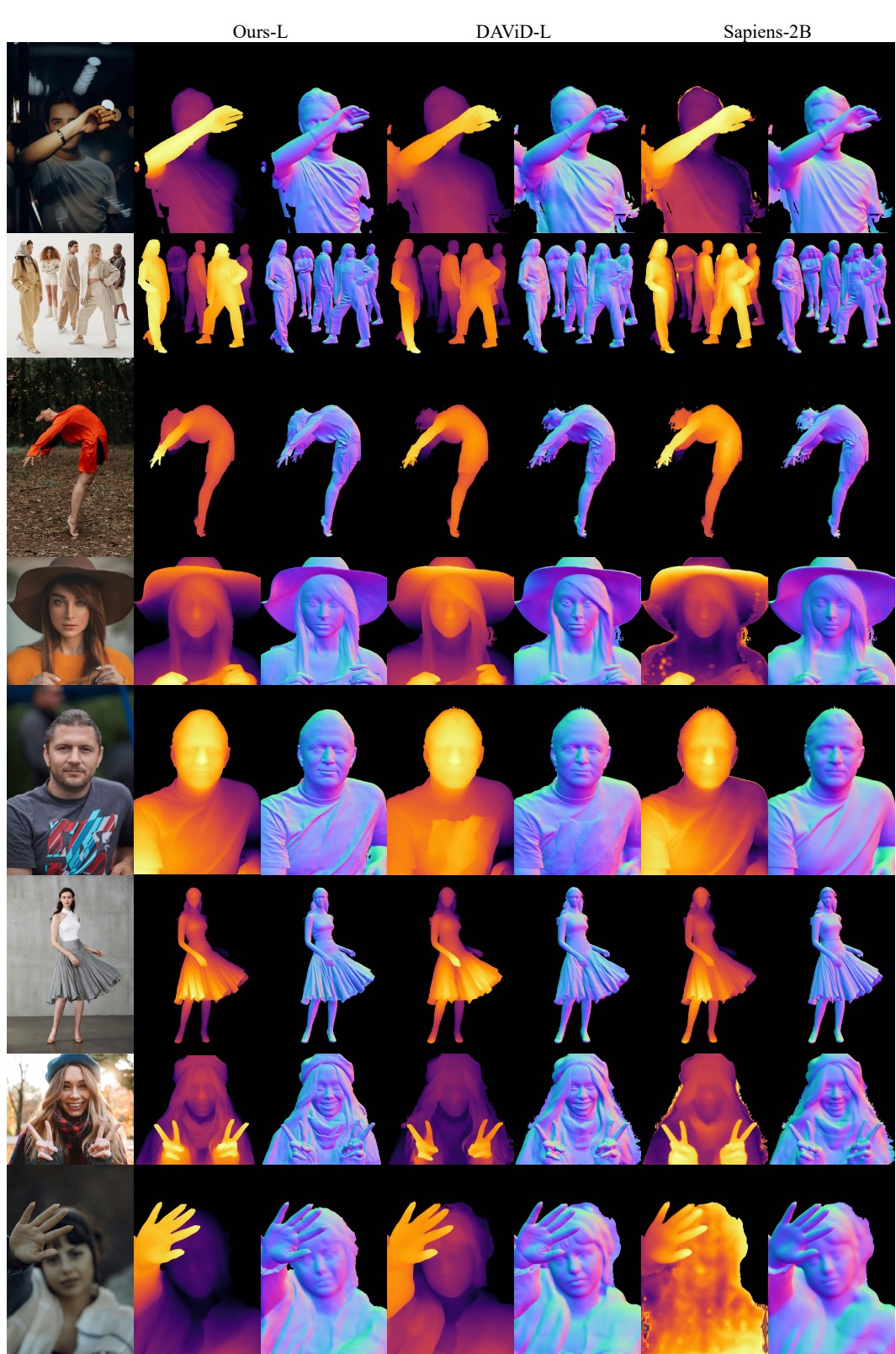

Figure 9: Additional qualitative comparison on challenging images in the wild.

