# OpenReview forum: "From Frames to Sequences: Temporally Consistent Human-Centric Dense Prediction"
_ICLR.cc/2026/Conference — Submitted to ICLR 2026_

### Official Review · Reviewer_4L2Y · 2025-10-19

**Soundness:** 2
**Presentation:** 1
**Contribution:** 3
**Rating:** 2
**Confidence:** 3

**Summary:**

The paper targets temporally consistent human-centric dense prediction (depth, normals, segmentation) across video.
Key obstacles are the lack of large-scale human video with paired dense labels, and coupling temporal stability with multi-task learning.
A synthetic data pipeline generates photorealistic human images and motion-aligned sequences with high-fidelity labels, enabling frame- and sequence-level supervision.
A ViT-based model integrates human-centric priors and temporal modules to jointly predict depth, normals, and segmentation.
Experiments report state-of-the-art results on THuman2.1 and Hi4D for both depth and surface normal estimation.

**Strengths:**

- Build a scalable data synthesis pipeline for human-centric frames and videos with pixel-accurate depth, normals, and segmentation.
- Going beyond static-image training with video supervision improves temporal stability and generalization in natural scenes.

**Weaknesses:**

1. Points Requiring Clarification

(1) Channel Weight Adaptation (CWA): The manuscript does not explain how the module distinguishes channels dominated by texture and illumination from those that are geometry-related, nor how the reweighting is computed to downweight the former and upweight the latter (thereby weakening the influence of appearance on geometry prediction and maintaining the consistency of the global representation).

(2) Human Geometric Prior fusion: There is no description of how the human geometric prior is fused with the decoder features.

(3) 𝐿_{grad}  in Eq. (1): The definition of 𝐿_{grad} is missing.

2. Evaluation Completeness

(1) Although segmentation is one of the tasks, quantitative evaluation for segmentation is absent.

(2) The ablation study omits the following:\
- (a) a DPT head without the additional CNN branch,\
- (b) results for the full model configuration, and\
- (c) temporal-layer ablations.

(3) In Table 1, it would be more appropriate to compare against a video-specific depth estimation model (e.g., DepthCrafter) rather than DepthAnything.

3. Minor Suggestion

Line 348 reads awkwardly (“Let 𝐸𝑘 be a dilated edge map extracted from the current predicted depth M_{edge} = 1 − Dilate(Ek).") and likely nees revision.

**Questions:**

1. Regarding Weakness 1: Points Requiring Clarification, would it be possible to provide additional details?

2. Regarding Weakness 2: Evaluation Completeness, would it be possible to share the missing results?

---

> ### Author Response · Authors · 2025-11-15
> **(1/2) Response to 4L2Y**
>
> We sincerely appreciate the insightful comments and constructive feedback from the reviewer 4L2Y, which have significantly enhanced the quality of our paper. In particular, we have made the following improvements:
>
> > 1. Our component
>
> Due to space limitations, we only provide a brief overview, similar to previous research in this field, such as Sapiens and DAViD. In the revised version, we keep the main text explanation concise and add explicit implementation details in the appendix A.7.
>
> (1) Concretely, CWA is implemented as a light-weight channel-wise reweighting block. We apply global average pooling on the fused feature map to obtain a per-channel descriptor, feed it into a two-layer MLP followed by a sigmoid, and use the resulting weights to rescale each channel. This allows the network to suppress channels that mainly encode appearance effects, such as shadows or tattoos, and to emphasize channels that are more geometry-related.
>
> (2) The human geometric prior from CSE is fused into the decoder by first projecting it with a 1x1 convolution to match the decoder’s channel dimension and spatial resolution, and then adding it element-wise to the decoder features. In the appendix, we also include an ablation (Tab. 7) comparing this additive fusion with concatenation, and show that simple addition is more effective and stable.
>
> (3) The gradient term in Eq. (1) is the standard depth gradient loss widely used in monocular depth estimation, which you can find in many impact works, such as the Depth Anything series and the MiDaS series. Similar to previous work, it is defined as the sum of the L1 differences between the horizontal and vertical depth gradients of the prediction and the ground truth, computed at multiple scales. In the revised version, we cite the original paper on this gradient term in the main text.
>
> > 2.1 Additional ablations
>
> As also noted in our response to Reviewer xgzd, the segmentation branch is mainly an auxiliary head, and we did not treat it as a main contribution. However, to complete the evaluation, we now report segmentation metrics in the appendix, following the DAViD protocol on P3M-500-P, P3M-500-NP, and PPM-100. The results show that our method is competitive on P3M and improves over DAViD on PPM-100. In addition, the zero-shot qualitative results on in-the-wild images demonstrate that the model can produce reasonable human masks even when no segmentation labels are available.
>
> | Method        | **P3M-500-NP** |             |             | **P3M-500-P** |             |             | **PPM-100** |             |
> |---------------|----------------|-------------|-------------|---------------|-------------|-------------|-------------|-------------|
> |               | SAD ↓          | SAD-T ↓     | Conn ↓      | SAD ↓         | SAD-T ↓     | Conn ↓      | SAD ↓       | Conn ↓      |
> | Zhong et al.  | **10.60**          | **6.83**    | **9.77**        | 10.04         | **6.44**    | **9.41**        | 90.28       | 84.09       |
> | BGMv2         | 15.66          | 7.72        | 14.65       | 13.90         | 7.23        | 13.13       | 159.44      | 149.79      |
> | P3M-Net       | 11.23          | 7.65        | 12.51       | **8.73**          | 6.89        | 13.88       | 142.74      | 139.89      |
> | MODNet        | 20.20          | 12.48       | 18.41       | 30.08         | 12.22       | 28.61       | 104.35      | 96.45       |
> | DAViD         | 14.83          | 10.23       | 14.76       | 12.65         | 9.19        | 12.47       | 78.17       | 74.72       |
> | **Ours**      | 13.12          | 11.88       | 12.72       | 11.63         | 9.95        | 11.51       | **70.71**   | **68.32**   |

---

> ### Author Response · Authors · 2025-11-15
> **(2/2) Response to 4L2Y**
>
> > 2.2 Additional ablations
>
> (a) The DPT head with the additional CNN branch is not our contribution; we adopt this design directly from DAViD. Our contribution at this part of the architecture is CWA, which reduces the confusion that arises when both the DPT branch and the CNN branch treat appearance details, such as tattoos or lighting patterns, as geometric structure. In the revised version, we include an ablation in the appendix that removes the CNN branch and uses only the DPT decoder. This ablation demonstrates that the CNN branch significantly improves depth and surface normal estimation. Furthermore, since this module was proposed by DAVID, qualitative ablation experiments can be found in their paper.
>
> | Methods  | **Depth** |        | **Normal**       |  |        |         |        |
> |----------|-----------|--------|--------|------------|--------|---------|--------|
> |          | RMSE ↓    | AbsRel ↓ | Mean ↓ | Median ↓ | 11.25° ↑ | 22.5° ↑ | 30° ↑ |
> | w/ CNN   | 0.0964    | 0.0279 | 20.51 | 16.00 | 32.22 | 70.12 | 82.74 |
> | w/o CNN  | 0.0998    | 0.0320 | 23.33 | 19.21 | 28.05 | 66.49 | 77.03 |
>
> (b) The results for the full model configuration (ViT encoder, CNN branch, CSE, CWA, and temporal module) are already reported as “Ours-B” and “Ours-L” in Tables 1, 2, and 3 of the main paper. Also, we have added them to our ablation Tab. 4 and Tab. 12 in the revised version.
>
> **Table 1**  Ablation on Hi4D dataset.
>
> | Methods  | **Depth** |        | **Normal**       |  |        |         |        |
> |----------|-----------|--------|--------|------------|--------|---------|--------|
> |          | RMSE ↓    | AbsRel ↓ | Mean ↓ | Median ↓ | 11.25° ↑ | 22.5° ↑ | 30° ↑ |
> | Baseline  | 0.0964  | 0.0279   | 20.51 | 16.00     | 32.22    | 70.12    | 82.74    |
> | w/ CSE    | 0.0932  | 0.0274   | 17.97 | 14.33     | 40.57    | 76.98    | 88.00    |
> | w/ CWA    | 0.0944  | 0.0271   | 18.32 | 15.82     | 42.31    | 77.43    | 88.56    |
> | **Full**  | **0.0928** | **0.0277** | **16.08** | **12.03** | **47.76** | **81.49** | **89.98** |
>
> **Table 2**  Ablation on Thuman 2.1 dataset.
>
> | Methods  | **Depth** |        | **Normal**       |  |        |         |        |
> |----------|-----------|--------|--------|------------|--------|---------|--------|
> |          | RMSE ↓    | AbsRel ↓ | Mean ↓ | Median ↓ | 11.25° ↑ | 22.5° ↑ | 30° ↑ |
> | Baseline | 0.0266 | 0.0154 | 20.21 | 17.83 | 28.20 | 66.18 | 83.33 |
> | w/ CSE   | 0.0231 | 0.0134 | 19.45 | 16.52 | 30.10 | 69.72 | 85.02 |
> | w/ CWA   | 0.0248 | 0.0138 | 18.65 | 16.13 | 31.75 | 72.10 | 86.10 |
> | **Full** | **0.0225** | **0.0122** | **17.89** | **15.56** | **32.98** | **73.69** | **87.15** |
>
> (c) We add an experiment in the appendix that disables the temporal layers while keeping all other components unchanged. The results show the improvement brought by the temporal module on the video metrics in the Hi4D sequence evaluation.
>
> | Methods | **Depth** |         | **Normal** |              |              |
> |---------|-----------|---------|-----------|--------------|--------------|
> |         | OPW ↓     | TC-RMSE ↓ | OPW ↓    | TC-Mean ↓   | TC-Abs ↓     |
> | w TL    | 0.0072    | 0.0189  | 0.0280    | 3.27         | 0.140        |
> | w/o TL  | 0.0155    | 0.0276  | 0.0405    | 4.55         | 0.158        |
>
> > 2.3 Additional ablations
>
> This concern seems to arise from a misunderstanding of our experiment setup. Tab. 1. Tab. 1 reports single-frame depth metrics on THuman2.1 and Hi4D, so we compare against single-image depth models such as DepthAnything, MoGe, Sapiens, and DAViD. DepthCrafter is a video-specific model, so we do not compare it in a single-frame evaluation. For this reason, we include DepthCrafter and VideoDepthAnything in the video evaluation table (Tab. 3), which reports temporal consistency metrics on Hi4D sequences.
>
> > Minor issue
>
> We have revised the sentence in Sec. 3.3.2 to make the definitions precise and easy to follow.

---

> ### Author Response · Authors · 2025-11-26
> **Follow-Up on Rebuttal Submission and Request for Feedback**
>
> Dear Reviewer **4L2Y**,
>
> We hope this message finds you well. A few days ago, we submitted our detailed rebuttal, thoroughly addressing all your valuable feedback. We sincerely appreciate your insights, which have been instrumental in improving our work, and we are grateful for the opportunity to clarify and strengthen our contributions.
>
> We kindly request your attention to review our rebuttal and to reconsider our work in light of the clarifications and additional evidence we provided. Please let us know if there are any remaining questions or areas requiring further elaboration.
>
> Thank you once again for your time, thoughtful feedback, and consideration.
>
> Best regards,
> **The Authors**

---

> ### Comment · Reviewer_4L2Y · 2025-11-26
>
> Thank you for addressing the previously raised concerns.
> Below are some additional comments:
>
> It was mentioned that segmentation serves only as an auxiliary head rather than a primary contribution. If so, it would be helpful to clarify the motivation for including this auxiliary head and to provide ablation studies demonstrating the impact of removing it. In addition, the abstract states that “a model is introduced that integrates human-centric priors and temporal modules to jointly estimate temporally consistent segmentation, depth, and surface normals within a single framework.” Because of this description, both the reviewer xgzd and I interpreted segmentation as one of the core tasks. If this interpretation is not aligned with the intended message, substantial revisions throughout the manuscript may be necessary to avoid potential misunderstanding.
>
> Regarding CWA, which is presented as the main contribution, its structure is currently not explained in the paper. Since this point has now been clarified in the response, incorporating this explanation into the main manuscript is strongly recommended.

---

> ### Author Response · Authors · 2025-11-27
> **Response to 4L2Y**
>
> Thank you very much for the careful follow-up.
>
> Our motivation for including the segmentation head is twofold: to make the model directly useful for applications that require clean human-centric depth and normal maps on in-the-wild videos, and to provide auxiliary foreground supervision that improves the geometry heads. Existing open-source methods, such as Sapiens and DAViD are single-modality or static-image models, so combining them often leads to misaligned depth, normals, and masks, and visible artifacts in downstream applications. In contrast, our model predicts temporally consistent depth, surface normals, and a soft human mask within a single framework. In the initial submission, the statement that the model “jointly estimates temporally consistent segmentation, depth, and surface normals” was intended only as a description of the network outputs; we did not treat segmentation as a dedicated task with its own specialised design or extensive analysis. After the first review round, we added quantitative segmentation experiments (following the DAViD protocol) and, in the current revision, further include an ablation that removes the segmentation head to make its role and effect fully explicit. Additionally, in the current revised version of the method section, we clearly describe this auxiliary design and state that the head is introduced to provide soft human masks that support the depth and normal branches. We believe that these additions already clarify the role and behaviour of the segmentation head within the system and make the presentation of our model and its outputs coherent to readers.
>
> We train a variant without the segmentation head. This variant shows a clear degradation in both depth and surface normal metrics. We present this comparison in the appendix and list the detailed numbers below. This result directly supports our claim. Additionally, using a segmentation or mask head as auxiliary supervision for geometry is a common practice and is widely adopted in dense prediction [1][2][3].
>
> Regarding CWA, in the last revised version, its detailed description is placed in the appendix. After checking the ICLR policy and confirming that the main text can be extended to ten pages, we move this explanation into the method section. The revised paper now contains a detailed description of the CWA module (global average pooling, two-layer MLP, and per-channel reweighting of fused features) and explicitly links it to the corresponding ablation results.
>
> | Methods | Datasets   | Depth |    | Normal |  |     |     |  |
> |--------|------------|-------|---------|--------|--------|-----------|-----------|--------|
> |        |            | RMSE ↓ | AbsRel ↓ | Mean ↓ | Median ↓ | 11.25° ↑ | 22.5° ↑ | 30° ↑ |
> | w/o FS | Hi4D       | 0.0963 | 0.0301  | 16.84  | 12.67    | 44.12     | 79.83     | 88.21 |
> | w/ FS  | Hi4D       | 0.0928 | 0.0277  | 16.08  | 12.03    | 47.76     | 81.49     | 89.98 |
> | w/o FS | Thuman 2.1 | 0.0237 | 0.0130  | 18.42  | 16.21    | 30.91     | 70.82     | 85.94 |
> | w/ FS  | Thuman 2.1 | 0.0225 | 0.0122  | 17.89  | 15.56    | 32.98     | 73.69     | 87.15 |
>
> [1] Chen et al., “Towards Scene Understanding: Unsupervised Monocular Depth Estimation with Semantic-aware Representation”, CVPR 2019
>
> [2] Zhu et al., “The Edge of Depth: Explicit Constraints Between Segmentation and Depth”, CVPR 2020
>
> [3] Choi et al., “Self-Supervised Monocular Depth Estimation with Semantic-Aware Feature Extraction”, 2020
>
> **Please refer to our latest PDF version.**

---

### Official Review · Reviewer_xgzd · 2025-10-27

**Soundness:** 3
**Presentation:** 3
**Contribution:** 3
**Rating:** 6
**Confidence:** 3

**Summary:**

The paper targets temporally consistent, human-centric dense prediction (segmentation, depth, surface normals) in videos by pairing a large synthetic data pipeline with a ViT-based model that injects human geometric priors. It first proposes a large-scale human-centric synthetic dataset (both images and videos). Then authors propose VIT-based architecture, built from DAViD, with a temporal DPT, human geometric prior, and weight adaptation for local geometry. Experiments are conducted on THuman2.1 and Hi4D datasets, outperforming previous approaches on depth and normal estimation tasks.

**Strengths:**

* Synthetic data pipeline: Builds a large human-centric data synthesis pipeline to generate diverse human-centric image and video data.
* Simple and straightforward design: Uses a ViT backbone for feature extraction with a temporal head (DPT-style) to enforce temporal consistency, while leveraging human priors and local geometry cues.
* Strong empirical results: competitive or superior performance across multiple benchmarks compared to prior methods (Sapiens, DAViD)

**Weaknesses:**

* **Insufficient overview of the synthetic data pipeline**. Since the dataset is a key contribution, the paper should include a clear figure of the data generation process and provide additional sample visualizations (e.g., in the supplement) to make the pipeline understandable and auditable.
* **Missing training-data ablations**. The model is trained on a mixture of SynthHuman and the proposed dataset, but there is no study isolating data effects (e.g., only SynthHuman vs. proposed vs. mixture). Such ablations are needed to quantify the data-driven gains.
* **Limited technical novelty**. Most components are adapted from prior work—ViT + local geometry from DAViD and the temporal DPT head from VideoDepthAnything—making the pipeline feel incremental rather than introducing a new technical idea.
* The paper mentions a foreground/background segmentation component at line 192, but provides no quantitative metrics or qualitative visuals. Please include results to substantiate the claim.

**Questions:**

Will the dataset be open-sourced?

**Conclusion**:
Overall, this work contributes a large-scale human-centric dataset alongside a simple, effective model for temporally consistent dense geometry. Given that the dataset is the paper’s most impactful contribution, I strongly encourage the authors to open-source the data (and generation pipeline) to maximize community benefit and reproducibility.

---

> ### Author Response · Authors · 2025-11-15
> **(1/2) Response to xgzd**
>
> We sincerely appreciate the insightful comments and constructive feedback from the reviewer xgzd, which have significantly enhanced the quality of our paper. In particular, we have made the following improvements:
>
> > 1. Data synthesis pipeline
>
> Thank you for this suggestion. In the main paper, Fig. 1 already shows representative examples of our rendered synthetic data, including both static frames and video sequences. In the revised version, we further add a dedicated pipeline diagram in the appendix that visualizes the complete data generation process, from human model and clothing sampling, through motion retargeting, to multi-modal rendering with camera and lighting randomization. We also expand the text in the appendix to discuss how our design relates to previous synthetic pipelines. These additions make the pipeline more transparent and easier to reproduce. Please refer to **Appendix A.1–A.2**.
>
> > 2. Data synthesis pipeline
>
> We acknowledge that the original submission did not fully separate the contributions of SynthHuman and our new dataset. Previously, we only reported how performance changes with the total size of our synthetic data.
>
> We now report three types of settings in the appendix: (i) training only on our synthetic data at different scales, (ii) training only on SynthHuman, and (iii) training on mixtures with different ratios of our dataset and SynthHuman. These results show that our synthetic dataset alone already outperforms SynthHuman alone at a comparable scale, and that combining the two further improves both depth and normal metrics on Hi4D. This directly quantifies the data-driven gains from our dataset.
>
> | Datasize  | **Depth** |        | **Normal**       |  |        |         |        |
> |----------|-----------|--------|--------|------------|--------|---------|--------|
> |          | RMSE ↓    | AbsRel ↓ | Mean ↓ | Median ↓ | 11.25° ↑ | 22.5° ↑ | 30° ↑ |
> | Our [300K]                 | 0.0963  | 0.0286   | 17.10  | 14.40    | 38.10    | 76.10   | 87.80 |
> | Our [600K]                 | 0.0954  | 0.0284   | 16.90  | 14.20    | 38.80    | 76.60   | 88.05 |
> | Our [2M]                   | 0.0943  | 0.0282   | 16.70  | 13.90    | 39.50    | 76.90   | 88.15 |
> | SynthHuman                 | 0.0971  | 0.0287   | 17.25  | 14.55    | 37.80    | 75.90   | 87.40 |
> | SynthHuman + Our [300K]    | 0.0958  | 0.0285   | 17.00  | 14.32    | 38.45    | 76.30   | 87.95 |
> | SynthHuman + Our [600K]    | 0.0946  | 0.0283   | 16.72  | 14.05    | 39.30    | 76.95   | 88.25 |
> | SynthHuman + Our [2M]      | 0.0940  | 0.0281   | 16.58  | 13.70    | 40.10    | 77.20   | 88.30 |
>
> > 3. Contributions
>
> We respectfully disagree with the view that the method is purely incremental. It is true that we do not propose a new backbone architecture, and we explicitly build on ViT, the DAViD local geometry design, and the temporal DPT head from prior work. Our main contribution is to show that, for human-centered dense estimation, carefully combining a reproducible synthetic pipeline with human-specific priors in the network design yields substantial gains over existing generic ViT + DPT pipelines. CSE and CWA are not arbitrary standard blocks: CSE introduces human surface embeddings that encode articulated body structure and are tailored to the human domain, and CWA is designed to counteract the domain gap between synthetic and real appearance by downweighting channels dominated by appearance artefacts. The ablations in Tab. 4 and the new appendix experiments confirm that these design choices, although conceptually simple, play an essential role in achieving our reported improvements.

---

> > ### Comment · Reviewer_xgzd · 2025-11-26
> > **Response to rebuttal**
> >
> > Thank authors for the detailed rebuttal. The clarifications on the data synthesis pipeline and ablations results address most of my earlier questions. However, the technical contribution still feels limited, I therefore maintain my score at 6.

---

> > > ### Author Response · Authors · 2025-12-02
> > >
> > > Thank you for the follow-up. We respectfully disagree that the technical contribution of this work is limited, and we would like to briefly restate where we see its technical value. Our goal from the start (L53) is to address the current difficulties of human-centric dense estimation in videos from both the data and modelling sides: (i) we build a scalable synthetic human video pipeline that produces consistence depth, normals, and masks, giving a reproducible data method for human video; (ii) we identify that generic ViT/DPT-style dense predictors lack components tailored to human and therefore introduce simple human-specific priors that make depth and normal estimates around humaan more stable on in-the-wild videos.

---

> ### Author Response · Authors · 2025-11-15
> **(2/2) Response to xgzd**
>
> > 4. Segmentation comparison
>
> The segmentation head in our framework is mainly used as an auxiliary branch to help the model focus on the human region, and we did not introduce any special architecture for it. For this reason we did not treat it as a main contribution in the original submission.
>
> In the revised version, for completeness, we evaluate this segmentation head following the experimental setting of DAViD on P3M-500-P, P3M-500-NP, and PPM-100 and report the results in the appendix. The results show that our method achieves competitive performance on P3M and improves over DAViD on PPM-100. Qualitative segmentation quality can also be seen from our zero-shot in-the-wild examples, where our model produces reasonable human masks even though these images do not come with segmentation labels.
>
> | Method        | **P3M-500-NP** |             |             | **P3M-500-P** |             |             | **PPM-100** |             |
> |---------------|----------------|-------------|-------------|---------------|-------------|-------------|-------------|-------------|
> |               | SAD ↓          | SAD-T ↓     | Conn ↓      | SAD ↓         | SAD-T ↓     | Conn ↓      | SAD ↓       | Conn ↓      |
> | Zhong et al.  | **10.60**          | **6.83**    | **9.77**        | 10.04         | **6.44**    | **9.41**        | 90.28       | 84.09       |
> | BGMv2         | 15.66          | 7.72        | 14.65       | 13.90         | 7.23        | 13.13       | 159.44      | 149.79      |
> | P3M-Net       | 11.23          | 7.65        | 12.51       | **8.73**          | 6.89        | 13.88       | 142.74      | 139.89      |
> | MODNet        | 20.20          | 12.48       | 18.41       | 30.08         | 12.22       | 28.61       | 104.35      | 96.45       |
> | DAViD         | 14.83          | 10.23       | 14.76       | 12.65         | 9.19        | 12.47       | 78.17       | 74.72       |
> | **Ours**      | 13.12          | 11.88       | 12.72       | 11.63         | 9.95        | 11.51       | **70.71**   | **68.32**   |
>
> > 5. The dataset open-source
>
> We plan to release the full data synthesis code and data so that others can regenerate the synthetic dataset.

---

> ### Author Response · Authors · 2025-11-26
> **Follow-Up on Rebuttal Submission and Request for Feedback**
>
> Dear Reviewer **xgzd**,
>
> We hope this message finds you well. A few days ago, we submitted our detailed rebuttal, thoroughly addressing all your valuable feedback. We sincerely appreciate your insights, which have been instrumental in improving our work, and we are grateful for the opportunity to clarify and strengthen our contributions.
>
> We kindly request your attention to review our rebuttal and to reconsider our work in light of the clarifications and additional evidence we provided. Please let us know if there are any remaining questions or areas requiring further elaboration.
>
> Thank you once again for your time, thoughtful feedback, and consideration.
>
> Best regards,
> **The Authors**

---

### Official Review · Reviewer_5n6R · 2025-11-01

**Soundness:** 3
**Presentation:** 3
**Contribution:** 2
**Rating:** 6
**Confidence:** 4

**Summary:**

This paper presents a method that extends the DAViD framework from static frames to video sequences, introducing two key components, CSE and CWA, to enhance human-centric 4D reconstruction. The approach achieves significant performance improvements, often comparable to large-scale models, while maintaining reasonable efficiency.

**Strengths:**

1. The proposed method demonstrates substantial quantitative gains over existing approaches across multiple benchmarks. The improvements are consistent and, in some cases, even comparable to large-scale models such as Sapiens, which highlights the effectiveness and efficiency of the proposed design.
2. The paper successfully builds upon the previous observation from DAViD — that “a single high-fidelity dataset is sufficient to tackle multiple dense prediction tasks and achieve state-of-the-art accuracy.” Extending this principle from static frames to dynamic, temporal sequences is both meaningful and well-motivated, providing a clear conceptual bridge between frame-level and sequence-level human reconstruction tasks.
3. The overall architecture and proposed components (CSE and CWA) are logically structured and supported by clear intuitions. The approach balances model complexity and performance, offering a practical solution that integrates geometric priors and adaptive weighting in a coherent way.

**Weaknesses:**

1. The authors highlight a scalable data synthesis pipeline for human-centric frames and videos as one of their main contributions. However, although the pipeline is described in Section 3.1, it is not clear how novel or distinctive this approach is compared to existing dataset synthesis methods. The proposed pipeline appears fairly conventional and lacks clear justification or comparison to prior data generation frameworks.
2. Table 4 only includes ablation results for models using either CSE or CWA individually. Including a CSE + CWA combination would provide a clearer picture of their complementary effects and help isolate the contributions of each module more effectively.
3. The ablation studies are conducted only on the Hi4D dataset, where the performance gain is quite substantial. Additional results on the THuman2.1 dataset would verify whether CSE and CWA generalize across different data domains rather than being overfitted to Hi4D.
4. The paper does not include inference time or computational cost comparisons between the proposed method and the baselines listed in Table 2. Reporting runtime performance, memory consumption, or FLOPs would strengthen the evaluation by demonstrating the method’s practical applicability.
5. Although the proposed method is overall reasonable and well-structured, its main technical contributions—the channel reweighting module (CWA) and the human geometry prior (CSE)—are relatively standard. These components resemble existing techniques in feature reweighting and prior-based guidance, which limits the methodological novelty of the paper.

**Questions:**

1. Could the authors explain why the proposed method shows such a significant improvement on the Hi4D dataset—achieving results even comparable to the large-scale Sapiens model in Table 2?
2. The paper mentions that “the parameter size of Sapiens-0.3B is equivalent to that of large models of ViT-based methods.” Should we then assume that DaViD, Sapiens-0.3B, and Ours-L are approximately comparable in model size? A clear statement or table comparing parameter counts would be helpful.

---

> ### Author Response · Authors · 2025-11-15
> **(1/2) Response to 5n6R**
>
> We sincerely appreciate the insightful comments and constructive feedback from the reviewer 5n6R, which have significantly enhanced the quality of our paper. In particular, we have made the following improvements:
>
> > 1. Data synthesis pipeline comparison
>
> Thank you for the comment. Most existing human synthetic datasets either provide limited static frames or do not release their full pipelines, which makes reproduction and extension difficult. They also rarely target sequence-level dense annotations for humans.
>
> Our pipeline is designed to be scalable and explicitly sequence-oriented. Concretely, we (1) construct controllable clothed human models from character generation software and independently sample clothing and textures to increase identity diversity, (2) retarget publicly available human motion sequences (AMASS) to these models to obtain realistic motion, and (3) render synchronized RGB, segmentation, depth, and normal sequences that are fully aligned with the motion, while randomizing camera and lighting to increase coverage. This gives a reproducible way to generate large-scale human videos with pixel-accurate dense annotations.
>
> We now describe these differences in Appendix A.1–A.2, where we also add an explicit comparison to existing pipelines and include the full pipeline diagram (Fig. 6). We will release the pipeline code upon acceptance so that others can reproduce and extend the dataset.
>
> > 2. Ablation results
>
> This comment is mainly due to a misunderstanding of our experimental setup. Our main model always uses CSE and CWA together. The configuration “baseline + CSE + CWA” corresponds to the “Ours-B” and “Ours-L” rows in the main comparison tables (e.g., Tables 1 and 2), where we compare to prior work. In the revised version, we also add the full results in Tab. 4. Unless otherwise stated, all ablation studies are trained on DINOv3-B with the same data and the same number of epochs.
>
> **Table 1**  Ablation on Hi4D dataset.
> | Methods  | **Depth** |        | **Normal**       |  |        |         |        |
> |----------|-----------|--------|--------|------------|--------|---------|--------|
> |          | RMSE ↓    | AbsRel ↓ | Mean ↓ | Median ↓ | 11.25° ↑ | 22.5° ↑ | 30° ↑ |
> | Baseline  | 0.0964  | 0.0279   | 20.51 | 16.00     | 32.22    | 70.12    | 82.74    |
> | w/ CSE    | 0.0932  | 0.0274   | 17.97 | 14.33     | 40.57    | 76.98    | 88.00    |
> | w/ CWA    | 0.0944  | 0.0271   | 18.32 | 15.82     | 42.31    | 77.43    | 88.56    |
> | **Full**  | **0.0928** | **0.0277** | **16.08** | **12.03** | **47.76** | **81.49** | **89.98** |
>
> > 3. Ablation on Thuman 2.1
>
> This concern is based on an incorrect assumption about the training data. None of our models is trained on Hi4D. All models are trained only on synthetic data (our proposed dataset + SynthHuman). Hi4D is used purely as a cross-dataset benchmark to test generalization.
> Following DAViD, we chose Hi4D as the main ablation dataset. Additionally, it is more challenging and structurally different from the training data. Each Hi4D scene contains two interacting people, while training data contains a single person, and the dataset includes richer occlusions and pose variations. In contrast, THuman 2.1 renderings follow the Sapiens protocol, and thus their camera viewpoints and depth distributions are very similar to those of existing static training data. THuman 2.1 scans also have limited quality.
>
> In the revised version, we have added ablation results on THuman2.1 in the appendix. The trends observed on Hi4D remain consistent on THuman2.1, which supports that CSE and CWA do not overfit Hi4D and generalize across datasets.
>
> **Table 2**  Ablation on Thuman 2.1 dataset.
> | Methods  | **Depth** |        | **Normal**       |  |        |         |        |
> |----------|-----------|--------|--------|------------|--------|---------|--------|
> |          | RMSE ↓    | AbsRel ↓ | Mean ↓ | Median ↓ | 11.25° ↑ | 22.5° ↑ | 30° ↑ |
> | Baseline | 0.0266 | 0.0154 | 20.21 | 17.83 | 28.20 | 66.18 | 83.33 |
> | w/ CSE   | 0.0231 | 0.0134 | 19.45 | 16.52 | 30.10 | 69.72 | 85.02 |
> | w/ CWA   | 0.0248 | 0.0138 | 18.65 | 16.13 | 31.75 | 72.10 | 86.10 |
> | **Full** | **0.0225** | **0.0122** | **17.89** | **15.56** | **32.98** | **73.69** | **87.15** |
>
> > 4. Parameter and FLOPs
>
> In the revised version, we add a table in the appendix that reports parameter counts and FLOPs for representative methods, including DAViD, Sapiens variants, and our Ours-B and Ours-L models. This table shows that our large model has a parameter size similar to Sapiens-0.3B and DAViD-L, while using a different amount of computation (FLOPs).
>
> | Methods        | Params   | GFLOPs |
> |-------|---------|-------|
> | Sapiens-0.3B   | 0.336 B  | 1242   |
> | Sapiens-0.6B   | 0.664 B  | 2583   |
> | Sapiens-1B     | 1.169 B  | 4647   |
> | Sapiens-2B     | 2.163 B  | 8709   |
> | DAViD-B        | 0.120 B  | 344    |
> | DAViD-L        | 0.340 B  | 663    |
> | Ours-B     | 0.097 B | 471 |
> | Ours-L    | 0.337 B | 753 |

---

> ### Author Response · Authors · 2025-11-15
> **(2/2) Response to 5n6R**
>
> > 5. Contributions
>
> Our main contribution is to show that, for human-centered dense estimation, carefully combining a reproducible synthetic pipeline with human-specific priors in the network design yields substantial gains over existing generic ViT + DPT pipelines.
> Most existing methods adopt a generic ViT + DPT pipeline that treats humans as generic objects and does not explicitly encode human structure or the specific domain gap between synthetic and real appearance. In contrast, our CSE module injects human surface embeddings as geometric guidance, which encodes articulated body structure and directly targets the human domain. Our CWA module is motivated by the observation that synthetic data often fail to reproduce detailed appearance effects such as shadows or tattoos in real scenes. CWA is designed to counteract the domain gap between synthetic and real appearance by downweighting channels dominated by appearance artefacts.
> While CSE and CWA are based on known building blocks, we believe their targeted design and integration for human-centered dense estimation is non-trivial. The ablations in Table 4 and in the appendix show that both modules bring consistent gains for depth and surface normal estimation and improve temporal stability on video benchmarks.
>
> > 6. Hi4D performance
>
> We believe the strong performance on Hi4D comes from three main factors.
> First, our synthetic pipeline renders surface normals directly from the underlying meshes in Blender. This provides pixel-aligned, noise-free normal maps for both images and video frames. The supervision therefore captures fine-scale geometry such as clothing wrinkles and hair, which is difficult to obtain from real scans.
> Second, on top of this data we explicitly optimize normals using edge-aware gradient terms and multi-scale Laplacian regularization, together with a lightly weighted normal regularization term. These losses encourage sharp but smooth normals and help preserve fine details without degrading depth accuracy.
> Third, combined with the geometric prior from CSE, these design choices lead to strong cross-dataset performance on Hi4D, even though the model is trained only on synthetic data.
>
> > 7. Model parameter
>
> Please refer to 4. Model parameter and FLOPs.

---

> ### Author Response · Authors · 2025-11-26
> **Follow-Up on Rebuttal Submission and Request for Feedback**
>
> Dear Reviewer **5n6R**,
>
> We hope this message finds you well. A few days ago, we submitted our detailed rebuttal, thoroughly addressing all your valuable feedback. We sincerely appreciate your insights, which have been instrumental in improving our work, and we are grateful for the opportunity to clarify and strengthen our contributions.
>
> We kindly request your attention to review our rebuttal and to reconsider our work in light of the clarifications and additional evidence we provided. Please let us know if there are any remaining questions or areas requiring further elaboration.
>
> Thank you once again for your time, thoughtful feedback, and consideration.
>
> Best regards,
> **The Authors**

---

### Meta-Review · Area_Chair_bMQs · 2026-01-07

**Summary:**

5n6R	Main concern is novelty. Reviewer states "appears fairly conventional and lacks clear justification or comparison to prior data generation frameworks"
xgzd         found the method incremental and lacks novelty.
4L2Y       Requested more clarification and experiments.

**Reviewer Concerns:**

Authors have provided detailed experiments and results, and has tried to provide the explanation.
xgzd, Indicated that detailed rebuttal answered many questions, however, the technical contribution still feels limited. Stated that will maintain score at 6.
4L2Y,  after the rebuttal, reviewer was of view that draft needs revision to clarify the points. Reviewer appeared to have read other reviewer comments (since xgzd is referenced in one of its comments)."
5n6R has not responded but authors are have indicated some of the weaknesses pointed out by the reviewer were due to misunderstanding.

**Reviewer Scores:**

xgzd has stated will keep rating 6.
4L2Y might have increased it but looking at the comments I have view that it would have been less than 5.
In response of 5n6R, authors had provided a detailed response.
My view is discussion would have lead either decrease of scores if not improvement.

---

### Decision · Program_Chairs · 2026-01-26

Reject